# Private Stochastic Convex Optimization with Optimal Rates

**Raef Bassily**[*]
The Ohio State University
`bassily.1@osu.edu`

**Vitaly Feldman**[*]
Google Research. Brain Team.

**Kunal Talwar**[*]
Google Research. Brain Team.
`kunal@google.com`

**Abhradeep Thakurta**[*]
University of California Santa Cruz
Google Research. Brain Team.

## Abstract

We study differentially private (DP) algorithms for stochastic convex optimization (SCO). In this problem the goal is to approximately minimize the population loss given i.i.d. samples from a distribution over convex and Lipschitz loss functions. A long line of existing work on private convex optimization focuses on the empirical loss and derives asymptotically tight bounds on the excess empirical loss. However a significant gap exists in the known bounds for the population loss.

We show that, up to logarithmic factors, the optimal excess population loss for DP algorithms is equal to the larger of the optimal non-private excess population loss, and the optimal excess empirical loss of DP algorithms. This implies that, contrary to intuition based on private ERM, private SCO has asymptotically the same rate of $1/\sqrt{n}$ as non-private SCO in the parameter regime most common in practice. The best previous result in this setting gives rate of $1/n^{1/4}$. Our approach builds on existing differentially private algorithms and relies on the analysis of algorithmic stability to ensure generalization.

## 1 Introduction

Many fundamental problems in machine learning reduce to the problem of minimizing the expected loss (a.k.a. population loss) $\mathcal{L}(\mathbf{w}) = \mathbb{E}_{z \sim \mathcal{D}} [\ell(\mathbf{w}, z)]$ for convex loss functions of $\mathbf{w}$ given access to samples $z_1, \ldots, z_n$ from the data distribution $\mathcal{D}$. This problem arises in various settings, such as estimating the mean of a distribution, least squares regression, or minimizing a convex surrogate loss for a classification problem. This problem is commonly referred to as *Stochastic Convex Optimization* (SCO) and has been the subject of extensive study in machine learning and optimization [SSBD14]. In this work we study this problem with the additional constraint of differential privacy.

A closely related problem is that of minimizing the loss $\widehat{\mathcal{L}}(\mathbf{w}) = \frac{1}{n} \sum_i \ell(\mathbf{w}, z_i)$ on the sampled set of functions, often known as Empirical Risk Minimization (ERM). The problem of private ERM has been well-studied and tight upper and lower bounds are known for private ERM. We give nearly tight upper and lower bounds on the *excess population loss* (a.k.a. excess population risk). At first glance these two problems may appear to be essentially the same as an optimal algorithm for minimizing the empirical risk should also achieve the best bounds for the population risk itself, i.e. the best approach to private SCO is to use the best private ERM.

This simple intuition is unfortunately false, even in the non-private case. A natural approach of bounding the population loss is by proving an upper bound on $\mathbb{E}_{z_1, \ldots, z_n}[\sup_{\mathbf{w}}(\mathcal{L}(\mathbf{w}) - \widehat{\mathcal{L}}(\mathbf{w}))]$. This

---

[*]Part of this work was done while visiting the Simons Institute for the Theory of Computing.

is known as *uniform convergence*. There are examples of distributions over losses where uniform convergence based bounds are provably sub-optimal. For example, for convex Lipschitz losses in $d$-dimensional Euclidean space, the best bound on the population loss achievable via uniform convergence is $\Omega(\sqrt{d/n})$ [Fel16]. In contrast, SGD is known to achieve excess loss of $O(\sqrt{1/n})$ which is independent of the dimension. As a result, in the high-dimensional settings often considered in modern ML (when $n = \Theta(d)$), the optimal achievable excess loss is $O(\sqrt{1/n})$, whereas the uniform convergence bound is $\Omega(1)$.

This discrepancy implies that using private ERM and appealing to uniform convergence will not lead to optimal bounds for private SCO. The first work to address the population loss for private SCO is [BST14] which gives bounds based on several natural approaches. Their first approach is to use the generalization properties of differential privacy itself to bound the gap between the empirical and population losses [DFH+15, BNS+16], and thus derive bounds for SCO from bounds on ERM. This approach leads to a suboptimal bound for private SCO (specifically[2], $\approx \max\left(\frac{d^{\frac{1}{4}}}{\sqrt{n}}, \frac{\sqrt{d}}{\epsilon n}\right)$ [BST14, Sec. F]). For the important case of $d$ being on the order of $n$ and $\epsilon$ being on the order of one this results in $\Omega(n^{-\frac{1}{4}})$ bound on excess population loss. Their second approach uses stability induced by regularizing the empirical loss before it is minimized via a private ERM algorithm for strongly convex losses. This technique also yields a suboptimal bound on the excess population loss $\approx (d^{\frac{1}{4}}/\sqrt{\epsilon\, n})$.

There are two natural lower bounds that apply to private SCO. The lower bound of $\Omega(\sqrt{1/n})$ for the excess loss of non-private SCO applies for private SCO. Further it is not hard to show that lower bounds for private ERM translate to essentially the same lower bound for private SCO, leading to a lower bound of the form $\Omega(\frac{\sqrt{d}}{\epsilon n})$. We give a detailed argument for the lower bound in the full version [BFTT19]. In this work, we address the question:

*What is the optimal excess loss for private SCO? Is the rate of $O\left(\sqrt{\frac{1}{n}} + \frac{\sqrt{d}}{\epsilon n}\right)$ achievable?*

## 1.1 Our contribution

We show that the optimal rate of $O\left(\sqrt{\frac{1}{n}} + \frac{\sqrt{d}}{\epsilon n}\right)$ is achievable. In particular, we obtain the statistically optimal rate of $O(1/\sqrt{n})$ whenever $d = O(n)$. This is in contrast to the situation for private ERM where the cost of privacy grows with the dimension for all $n$.

In fact, under relatively mild smoothness assumptions, this rate is achieved by a variant of the standard noisy mini-batch SGD. The parameters of the scheme need to be tuned carefully to satisfy a delicate balance. The classical analyses for non-private SCO depend crucially on making only one pass over the dataset. However, a single pass noisy SGD is not sufficiently accurate as we need a non-trivial amount of noise in each step to carry out the privacy analysis. We rely instead on a different approach to generalization, known as *uniform stability* [BE02]. The stability parameter degrades with the number of passes over the dataset [HRS15, FV19], while the empirical accuracy improves as we make more passes. In addition, the batch size needs to be sufficiently large to ensure that the noise added for privacy is small. To satisfy all these constraints the parameters of the scheme need to be tuned carefully. Specifically we show that $\approx \min(n, n^2\epsilon^2/d)$ steps of SGD with a batch size of $\approx \max(\sqrt{\epsilon n}, 1)$ are sufficient to get all the desired properties.

Our second contribution is to show that the smoothness assumptions can be relaxed at essentially no additional loss. We use a general smoothing technique based on the Moreau-Yosida envelope operator that allows us to derive the same asymptotic bounds as the smooth case. This operator cannot be implemented efficiently in general, but for algorithms based on gradient steps we exploit the well-known connection between the gradient step on the smoothed function and the proximal step on the original function. Thus our algorithm is equivalent to (stochastic, noisy, mini-batch) proximal descent on the unsmoothed function. We show that our analysis in the smooth case is robust to inaccuracies in the computation of the gradient. This allows us to show that sufficient approximation to the proximal steps can be implemented in polynomial time given access to the gradient of the $\ell(w, z_i)$'s.

Finally, we show that *Objective Perturbation* [CMS11, KST12] also achieves optimal bounds for private SCO. However, objective perturbation is only known to satisfy privacy under some additional assumptions (most notably, Hessian being rank 1 on all points in the domain). The generalization analysis in this case is based on the uniform stability of the solution to strongly convex ERM. Aside from extending the analysis of this approach to population loss, we show that it can lead to algorithms for private SCO that use only near-linear number of gradient evaluations (whenever these assumptions hold). In particular, we give a variant of objective perturbation in conjunction with the stochastic variance reduced gradient descent (SVRG) with only $O(n \log n)$ gradient evaluations. We remark that the known lower bounds for uniform convergence [Fel16] hold even under those additional assumptions invoked in objective perturbation. Finding algorithms with near-linear running time in the general setting of SCO is a natural avenue for future research.

Our work highlights the importance of uniform stability as a tool for analysis of this important class of problems. We believe it should have applications to other differentially private statistical analyses.

**Related work:** Differentially private empirical risk minimization (ERM) is a well-studied area spanning over a decade [CM08, CMS11, JKT12, KST12, ST13, SCS13, DJW13, Ull15, JT14, BST14, TTZ15, STU17, WLK$^+$17, WYX17, INS$^+$19]. Aside from [BST14] and work in the local model of DP [DJW13], these works focus on achieving optimal *empirical* risk bounds under privacy. Our work builds heavily on algorithms and analyses developed in this line of work while contributing additional insights. Optimal bounds for private SCO are known for some simple subclasses of convex functions such as Generalized Linear Models [JT14, BST14] where uniform convergence bounds on the order of $1/\sqrt{n}$ are known [KST08].

## 2 Preliminaries

**Notation:** We use $\mathcal{W} \subset \mathbb{R}^d$ to denote the parameter space, which is assumed to be a convex, compact set. We denote by $M = \max_{\mathbf{w} \in \mathcal{W}} \|\mathbf{w}\|$ the $L_2$ radius of $\mathcal{W}$. We use $\mathcal{Z}$ to denote an arbitrary data universe and $\mathcal{D}$ to denote an arbitrary distribution over $\mathcal{Z}$. We let $\ell : \mathbb{R}^d \times \mathcal{Z} \to \mathbb{R}$ be a loss function that takes a parameter vector $\mathbf{w} \in \mathcal{W}$ and a data point $z \in \mathcal{Z}$ as inputs and outputs a real value.

The *empirical loss* of $\mathbf{w} \in \mathcal{W}$ w.r.t. loss $\ell$ and dataset $S = (z_1, \ldots, z_n)$ is defined as $\widehat{\mathcal{L}}(\mathbf{w}; \ S) \triangleq \frac{1}{n} \sum_{i=1}^n \ell(\mathbf{w}, z_i)$. The *excess empirical loss* of $\mathbf{w}$ is defined as $\widehat{\mathcal{L}}(\mathbf{w}; \ S) - \min_{\widetilde{\mathbf{w}} \in \mathcal{W}} \widehat{\mathcal{L}}(\widetilde{\mathbf{w}}; \ S)$.

The population loss of $\mathbf{w} \in \mathcal{W}$ with respect to a loss $\ell$ and a distribution $\mathcal{D}$ over $\mathcal{Z}$, is defined as $\mathcal{L}(\mathbf{w}; \mathcal{D}) \triangleq \mathbb{E}_{z \sim \mathcal{D}} [\ell(\mathbf{w}, z)]$. The *excess population loss* of $\mathbf{w}$ is defined as $\mathcal{L}(\mathbf{w}; \ \mathcal{D}) - \min_{\widetilde{\mathbf{w}} \in \mathcal{W}} \mathcal{L}(\widetilde{\mathbf{w}}; \mathcal{D})$.

**Definition 2.1** (Uniform stability). *Let $\alpha > 0$. A (randomized) algorithm $\mathcal{A} : \mathcal{Z}^n \to \mathcal{W}$ is $\alpha$-uniformly stable (w.r.t. loss $\ell : \mathcal{W} \times \mathcal{Z} \to \mathbb{R}$) if for any pair $S$, $S' \in \mathcal{Z}^n$ differing in at most one data point, we have*

$$\sup_{z \in \mathcal{Z}} \mathbb{E}_{\mathcal{A}} \left[ \ell\left(\mathcal{A}(S), z\right) - \ell\left(\mathcal{A}(S'), z\right) \right] \leq \alpha$$

*where the expectation is taken only over the internal randomness of $\mathcal{A}$.*

The following is a useful implication of uniform stability.

**Lemma 2.2** (See, e.g., [SSBD14]). *Let $\mathcal{A} : \mathcal{Z}^n \to \mathcal{W}$ be an $\alpha$-uniformly stable algorithm w.r.t. loss $\ell : \mathcal{W} \times \mathcal{Z} \to \mathbb{R}$. Let $\mathcal{D}$ be any distribution over $\mathcal{Z}$, and let $S \sim \mathcal{D}^n$. Then,*

$$\mathbb{E}_{S \sim \mathcal{D}^n, \mathcal{A}} \left[ \mathcal{L}\left(\mathcal{A}(S); \ \mathcal{D}\right) - \widehat{\mathcal{L}}\left(\mathcal{A}(S); \ S\right) \right] \leq \alpha.$$

**Definition 2.3** (Smooth function). *Let $\beta > 0$. A differentiable function $f : \mathbb{R}^d \to \mathbb{R}$ is $\beta$-smooth if for every $\mathbf{w}, \mathbf{v} \in \mathbb{R}^d$, we have*

$$f(\mathbf{v}) \leq f(\mathbf{w}) + \langle \nabla f(\mathbf{w}), \mathbf{v} - \mathbf{w} \rangle + \frac{\beta}{2} \|\mathbf{w} - \mathbf{v}\|^2.$$

In the sequel, whenever we attribute a property (e.g., convexity, Lipschitz property, smoothness, etc.) to a loss function $\ell$, we mean that for every data point $z \in \mathcal{Z}$, the loss $\ell(\cdot, z)$ possesses that property.

**Stochastic Convex Optimization (SCO):** Let $\mathcal{D}$ be an arbitrary (unknown) distribution over $\mathcal{Z}$, and $S = \{z_1, \ldots, z_n\}$ be a sample of i.i.d. draws from $\mathcal{D}$. Let $\ell : \mathcal{W} \times \mathcal{Z} \to \mathbb{R}$ be a convex loss

function. A (possibly randomized) algorithm for SCO uses the sample $S$ to generate an (approximate) minimizer $\widehat{\mathbf{w}}_S$ for $\mathcal{L}(\cdot;\ \mathcal{D})$. We measure the accuracy of $\mathcal{A}$ by the *expected* excess population loss of its output parameter $\widehat{\mathbf{w}}_S$, defined as:

$$\Delta\mathcal{L}\left(\mathcal{A};\ \mathcal{D}\right) \triangleq \mathbb{E}\left[\mathcal{L}(\widehat{\mathbf{w}}_S;\ \mathcal{D}) - \min_{\mathbf{w}\in\mathcal{W}}\mathcal{L}(\mathbf{w};\ \mathcal{D})\right],$$

where the expectation is taken over the choice of $S \sim \mathcal{D}^n$, and any internal randomness in $\mathcal{A}$.

**Differential privacy [DMNS06, DKM$^+$06]:** A randomized algorithm $\mathcal{A}$ is $(\epsilon, \delta)$-differentially private if, for any pair of datasets $S$ and $S'$ differ in exactly one data point, and for all events $\mathcal{O}$ in the output range of $\mathcal{A}$, we have

$$\mathbb{P}\left[\mathcal{A}(S) \in \mathcal{O}\right] \leq e^\epsilon \cdot \mathbb{P}\left[\mathcal{A}(S') \in \mathcal{O}\right] + \delta,$$

where the probability is taken over the random coins of $\mathcal{A}$. For meaningful privacy guarantees, the typical settings of the privacy parameters are $\epsilon < 1$ and $\delta \ll 1/n$.

**Differentially Private Stochastic Convex Optimization (DP-SCO):** An $(\epsilon, \delta)$-DP-SCO algorithm is a SCO algorithm that satisfies $(\epsilon, \delta)$-differential privacy.

## 3 Private SCO via Mini-batch Noisy SGD

In this section, we consider the setting where the loss $\ell$ is convex, Lipschitz, and smooth. We give a technique that is based on a mini-batch variant of Noisy Stochastic Gradient Descent (NSGD) algorithm [BST14, ACG$^+$16].

---

**Algorithm 1** $\mathcal{A}_{\mathsf{NSGD}}$: Mini-batch noisy SGD for convex, smooth losses

---

**Input:** Private dataset: $S = (z_1, \ldots, z_n) \in \mathcal{Z}^n$, $L$-Lipschitz, $\beta$-smooth, convex loss function $\ell$, convex set $\mathcal{W} \subseteq \mathbb{R}^d$, step size $\eta$, mini-batch size $m$, # iterations $T$, privacy parameters $\epsilon \leq 1$, $\delta \leq 1/n^2$.

1: Set noise variance $\sigma^2 := \frac{8T\,L^2\,\log(1/\delta)}{n^2\epsilon^2}$.
2: Set mini-batch size $m := \max\left(n\,\sqrt{\frac{\epsilon}{4T}},\ 1\right)$.
3: Choose arbitrary initial point $\mathbf{w}_0 \in \mathcal{W}$.
4: **for** $t = 0$ to $T-1$ **do**
5:     Sample a batch $B_t = \{z_{i_{(t,1)}}, \ldots, z_{i_{(t,m)}}\} \leftarrow S$ uniformly with replacement.
6:     $\mathbf{w}_{t+1} := \mathsf{Proj}_{\mathcal{W}}\left(\mathbf{w}_t - \eta \cdot \left(\frac{1}{m}\sum_{j=1}^m \nabla\ell(\mathbf{w}_t, z_{i_{(t,j)}}) + \mathbf{G}_t\right)\right)$, where $\mathsf{Proj}_{\mathcal{W}}$ denotes the Euclidean projection onto $\mathcal{W}$, and $\mathbf{G}_t \sim \mathcal{N}\left(\mathbf{0}, \sigma^2\mathbb{I}_d\right)$ drawn independently each iteration.
7: **return** $\overline{\mathbf{w}}_T = \frac{1}{T}\sum_{t=1}^T \mathbf{w}_t$

---

**Theorem 3.1** (Privacy guarantee of $\mathcal{A}_{\mathsf{NSGD}}$). *Algorithm 1 is $(\epsilon, \delta)$-differentially private.*

*Proof.* The proof follows from [ACG$^+$16, Theorem 1]. $\qquad\square$

The population loss attained by $\mathcal{A}_{\mathsf{NSGD}}$ is given by the next theorem.

**Theorem 3.2** (Excess population loss of $\mathcal{A}_{\mathsf{NSGD}}$). *Let $\mathcal{D}$ be any distribution over $\mathcal{Z}$, and let $S \sim \mathcal{D}^n$. Suppose $\beta \leq \frac{L}{M} \cdot \min\left(\sqrt{\frac{n}{2}},\ \frac{\epsilon\,n}{2\sqrt{2d\log(1/\delta)}}\right)$. Let $T = \min\left(\frac{n}{8},\ \frac{\epsilon^2\,n^2}{32\,d\,\log(1/\delta)}\right)$ and $\eta = \frac{M}{L\,\sqrt{T}}$. Then,*

$$\Delta\mathcal{L}\left(\mathcal{A}_{\mathsf{NSGD}};\ \mathcal{D}\right) \leq 10\,M\,L \cdot \max\left(\frac{\sqrt{d\,\log(1/\delta)}}{\epsilon\,n},\ \frac{1}{\sqrt{n}}\right)$$

Before proving the above theorem, we first state and prove the following useful lemmas.

**Lemma 3.3.** *Let $S \in \mathcal{Z}^n$. Suppose the parameter set $\mathcal{W}$ is convex and $M$-bounded. For any $\eta > 0$, the excess empirical loss of $\mathcal{A}_{\mathsf{NSGD}}$ satisfies*

$$\mathbb{E}\left[\widehat{\mathcal{L}}(\overline{\mathbf{w}}_T; S)\right] - \min_{\mathbf{w}\in\mathcal{W}} \widehat{\mathcal{L}}(\mathbf{w}; S) \leq \frac{M^2}{2\,\eta\,T} + \frac{\eta\,L^2}{2}\left(16\frac{T\,d\,\log(1/\delta)}{n^2\,\epsilon^2} + 1\right)$$

*where the expectation is taken with respect to the choice of the mini-batch (step 5) and the independent Gaussian noise vectors* $\mathbf{G}_1, \ldots, \mathbf{G}_T$.

*Proof.* The proof follows from the classical analysis of the stochastic oracle model (see, e.g., [SSBD14]). In particular, we can show that

$$\mathbb{E}\left[\widehat{\mathcal{L}}(\overline{\mathbf{w}}_T; S)\right] - \min_{\mathbf{w}\in\mathcal{W}} \widehat{\mathcal{L}}(\mathbf{w}; S) \leq \frac{M^2}{2\,\eta\,T} + \frac{\eta\,L^2}{2} + \eta\,\sigma^2\,d,$$

where the last term captures the extra empirical error due to privacy. The statement now follows from the setting of $\sigma^2$ in Algorithm 1. □

The following lemma is a simple extension of the results on uniform stability of GD methods that appeared in [HRS15] and [FV19, Lemma 4.3] to the case of *mini-batch noisy* SGD. We provide a proof for this lemma in the full version [BFTT19].

**Lemma 3.4.** *In* $\mathcal{A}_{\mathsf{NSGD}}$*, suppose* $\eta \leq \frac{2}{\beta}$*, where* $\beta$ *is the smoothness parameter of* $\ell$*. Then,* $\mathcal{A}_{\mathsf{NSGD}}$ *is* $\alpha$*-uniformly stable with* $\alpha = L^2\frac{T\,\eta}{n}$*.*

**Proof of Theorem 3.2**

By Lemma 2.2, $\alpha$-uniform stability implies that the expected generalization error is bounded by $\alpha$. Hence, by combining Lemma 3.3 with Lemma 3.4, we have

$$\mathbb{E}_{S\sim\mathcal{D}^n,\,\mathcal{A}_{\mathsf{NSGD}}}\left[\mathcal{L}(\overline{\mathbf{w}}_T;\,\mathcal{D})\right] - \min_{\mathbf{w}\in\mathcal{W}}\mathcal{L}(\mathbf{w};\,\mathcal{D}) \leq \mathbb{E}_{S\sim\mathcal{D}^n,\,\mathcal{A}_{\mathsf{NSGD}}}\left[\widehat{\mathcal{L}}(\overline{\mathbf{w}}_T;S)\right] - \min_{\mathbf{w}\in\mathcal{W}}\mathcal{L}(\mathbf{w};\,\mathcal{D}) + L^2\,\frac{\eta\,T}{n}$$

$$\leq \mathbb{E}_{S\sim\mathcal{D}^n,\,\mathcal{A}_{\mathsf{NSGD}}}\left[\widehat{\mathcal{L}}(\overline{\mathbf{w}}_T;S) - \min_{\mathbf{w}\in\mathcal{W}}\widehat{\mathcal{L}}(\mathbf{w};S)\right] + L^2\,\frac{\eta\,T}{n} \tag{1}$$

$$\leq \frac{M^2}{2\,\eta\,T} + \frac{\eta\,L^2}{2}\left(16\frac{T\,d}{n^2\,\epsilon^2} + 1\right) + L^2\,\frac{\eta\,T}{n}$$

where (1) follows from the fact that $\mathbb{E}_{S\sim\mathcal{D}^n}\left[\min_{\mathbf{w}\in\mathcal{W}}\widehat{\mathcal{L}}(\mathbf{w};S)\right] \leq \min_{\mathbf{w}\in\mathcal{W}}\mathbb{E}_{S\sim\mathcal{D}^n}\left[\widehat{\mathcal{L}}(\mathbf{w};S)\right] = \min_{\mathbf{w}\in\mathcal{W}}\mathcal{L}(\mathbf{w};\,\mathcal{D})$. Optimizing the above bound in $\eta$ and $T$ yields the values in the theorem statement for these parameters, as well as the stated bound on the excess population loss.

## 4 Private SCO for Non-smooth Losses

In this section, we consider the setting where the convex loss is non-smooth. First, we show a generic reduction to the smooth case by employing the smoothing technique known as *Moreau-Yosida regularization* (a.k.a. Moreau envelope smoothing) [Nes05]. Given an appropriately smoothed version of the loss, we obtain the optimal population loss w.r.t. the original non-smooth loss function. Computing the smoothed loss via this technique is generally computationally inefficient. Hence, we move on to describe a computationally efficient algorithm for the non-smooth case with essentially optimal population loss. Our construction is based on an adaptation of our noisy SGD algorithm $\mathcal{A}_{\mathsf{NSGD}}$ (Algorithm 1) that exploits some useful properties of Moreau-Yosida smoothing technique that stem from its connection to proximal operations.

**Definition 4.1** (Moreau envelope). *Let* $f : \mathcal{W} \rightarrow \mathbb{R}^d$ *be a convex function, and* $\beta > 0$*. The* $\beta$*-Moreau envelope of* $f$ *is a function* $f_\beta : \mathcal{W} \rightarrow \mathbb{R}^d$ *defined as*

$$f_\beta(\mathbf{w}) = \min_{\mathbf{v}\in\mathcal{W}}\left(f(\mathbf{v}) + \frac{\beta}{2}\|\mathbf{w} - \mathbf{v}\|^2\right), \quad \mathbf{w}\in\mathcal{W}.$$

Moreau envelope has direct connection with the proximal operator of a function defined below.

**Definition 4.2** (Proximal operator). *The prox operator of $f : \mathcal{W} \to \mathbb{R}^d$ is defined as*

$$\text{prox}_f(\mathbf{w}) = \arg\min_{\mathbf{v} \in \mathcal{W}} \left( f(\mathbf{v}) + \frac{1}{2}\|\mathbf{w} - \mathbf{v}\|^2 \right), \quad \mathbf{w} \in \mathcal{W}.$$

It follows that the Moreau envelope $f_\beta$ can be written as

$$f_\beta(\mathbf{w}) = f\left( \text{prox}_{f/\beta}(\mathbf{w}) \right) + \frac{\beta}{2}\|\mathbf{w} - \text{prox}_{f/\beta}(\mathbf{w})\|^2.$$

The following lemma states some useful, known properties of Moreau envelope.

**Lemma 4.3** (See [Nes05, Can11]). *Let $f : \mathcal{W} \to \mathbb{R}^d$ be a convex, L-Lipschitz function, and let $\beta > 0$. The $\beta$-Moreau envelope $f_\beta$ satisfies the following:*

1. $f_\beta$ *is convex, $2L$-Lipschitz, and $\beta$-smooth.*

2. $\forall \mathbf{w} \in \mathcal{W} \quad f_\beta(\mathbf{w}) \leq f(\mathbf{w}) \leq f_\beta(\mathbf{w}) + \frac{L^2}{2\beta}.$

3. $\forall \mathbf{w} \in \mathcal{W} \quad \nabla f_\beta(\mathbf{w}) = \beta\left(\mathbf{w} - \text{prox}_{f/\beta}(\mathbf{w})\right).$

Let $\ell : \mathcal{W} \times \mathcal{Z} \to \mathbb{R}$ be a convex, $L$-Lipschitz loss. For any $z \in \mathcal{Z}$, let $\ell_\beta(\cdot, z)$ denote the $\beta$-Moreau envelope of $\ell(\cdot, z)$. For a dataset $S = (z_1, \ldots, z_n) \in \mathcal{Z}^n$, let $\widehat{\mathcal{L}}_\beta(\cdot; S) \triangleq \frac{1}{n}\sum_{i=1}^{n} \ell_\beta(\cdot, z_i)$ be the empirical risk w.r.t. the $\beta$-smoothed loss. For any distribution $\mathcal{D}$, let $\mathcal{L}_\beta(\cdot; \mathcal{D}) \triangleq \mathbb{E}_{z \sim \mathcal{D}}\left[\ell(\cdot, z)\right]$ denote the corresponding population loss. The following theorem asserts that, with an appropriate setting for $\beta$, running $\mathcal{A}_{\text{NSGD}}$ over the $\beta$-smoothed losses $\ell_\beta(\cdot, z_i)$, $i \in [n]$ yields the optimal population loss w.r.t. the original non-smooth loss $\ell$.

**Theorem 4.4** (Excess population loss for non-smooth losses via smoothing). *Let $\mathcal{D}$ be any distribution over $\mathcal{Z}$. Let $S = (z_1, \ldots, z_n) \sim \mathcal{D}^n$. Let $\beta = \frac{L}{M} \cdot \min\left( \frac{\sqrt{n}}{4}, \frac{\epsilon\, n}{8\sqrt{d\,\log(1/\delta)}} \right)$. Suppose we run $\mathcal{A}_{\text{NSGD}}$ (Algorithm 1) over the $\beta$-smoothed version of $\ell$ associated with the points in $S$: $\{\ell_\beta(\cdot, z_i), i \in [n]\}$. Let $\eta$ and $T$ be set as in Theorem 3.2. Then, the excess population loss of the output of $\mathcal{A}_{\text{NSGD}}$ w.r.t. $\ell$ satisfies*

$$\Delta\mathcal{L}\left(\mathcal{A}_{\text{NSGD}}; \mathcal{D}\right) \leq 24\, M\, L \cdot \max\left( \frac{\sqrt{d\,\log(1/\delta)}}{\epsilon\, n}, \frac{1}{\sqrt{n}} \right)$$

*Proof.* Let $\overline{\mathbf{w}}_T$ be the output of $\mathcal{A}_{\text{NSGD}}$. Using property 1 of Lemma 4.3 together with Theorem 3.2, we have

$$\mathbb{E}_{S \sim \mathcal{D}^n, \mathcal{A}_{\text{NSGD}}}\left[\mathcal{L}_\beta(\overline{\mathbf{w}}_T; \mathcal{D})\right] - \min_{\mathbf{w} \in \mathcal{W}} \mathcal{L}_\beta(\mathbf{w}; \mathcal{D}) \leq 20\, M\, L \cdot \max\left( \frac{\sqrt{d\,\log(1/\delta)}}{\epsilon\, n}, \frac{1}{\sqrt{n}} \right).$$

By property 2 of Lemma 2 and the setting of $\beta$ in the theorem statement, for every $\mathbf{w} \in \mathcal{W}$, we have

$$\mathcal{L}_\beta(\mathbf{w}; \mathcal{D}) \leq \mathcal{L}(\mathbf{w}; \mathcal{D}) \leq \mathcal{L}_\beta(\mathbf{w}; \mathcal{D}) + 2\, M\, L \cdot \max\left( \frac{1}{\sqrt{n}}, \frac{2\sqrt{d\,\log(1/\delta)}}{\epsilon\, n} \right).$$

Putting these together gives the stated result. $\qquad\square$

**Computationally efficient algorithm $\mathcal{A}_{\text{ProxGD}}$ (NSGD + Prox)**

Computing the Moreau envelope of a function is computationally inefficient in general. However, by property 3 of Lemma 4.3, we note that evaluating the gradient of Moreau envelope at any point can be attained by evaluating the proximal operator of the function at that point. Evaluating the proximal operator is equivalent to minimizing a strongly convex function (see Definition 4.2). This can be approximated efficiently, e.g., via gradient descent. Since our $\mathcal{A}_{\text{NSGD}}$ algorithm (Algorithm 1) requires only sufficiently accurate gradient evaluations, we can hence use an efficient, approximate proximal operator to approximate the gradient of the smoothed losses. The gradient evaluations in

$\mathcal{A}_{\mathsf{NSGD}}$ will thus be replaced with such approximate gradients evaluated via the approximate proximal operator. The resulting algorithm, referred to as $\mathcal{A}_{\mathsf{ProxGD}}$, will approximately minimize the smoothed empirical loss without actually computing the smoothed losses.

Our construction of $\mathcal{A}_{\mathsf{ProxGD}}$ involves $\approx n^2 \cdot T^2 \cdot m$ gradient evaluations (of individual losses), where $T$ is the number of iterations of $\mathcal{A}_{\mathsf{NSGD}}$ reported in Theorem 3.2, and $m$ is its mini-batch size.

We argue that the approximate proximal operation will essentially have no impact on the guarantees of $\mathcal{A}_{\mathsf{ProxGD}}$ as compared to those of $\mathcal{A}_{\mathsf{NSGD}}$. In particular, in terms of *privacy*, the sensitivity of the approximate gradients (evaluated via the approximate prox operator) will basically remain the same as that of the exact gradients. In terms of *empirical error*, since the approximation error in the prox operations can be made sufficiently small (while maintaining computational efficiency), the impact of the approximation error on the empirical loss guarantee of $\mathcal{A}_{\mathsf{ProxGD}}$ will be negligible. Finally, in terms of *uniform stability*, again since the approximation error is sufficiently small, the error accumulated across iterations will have no pronounced impact on the uniform stability of $\mathcal{A}_{\mathsf{NSGD}}$ (established in Lemma 3.4). Putting these together shows that $\mathcal{A}_{\mathsf{ProxGD}}$ achieves the optimal population loss bound in Theorem 4.4.

A more detailed description of $\mathcal{A}_{\mathsf{ProxGD}}$ and its guarantees is given in the full version [BFTT19].

## 5 Private SCO via Objective Perturbation

In this section, we show that the technique known as objective perturbation [CMS11, KST12] can be used to attain optimal *population* loss under additional assumptions on the loss. These assumptions are invoked to ensure differential privacy. The excess *empirical* loss of this technique for smooth convex losses was originally analyzed in the aforementioned works, and was shown to be optimal by the lower bound in [BST14]. We revisit this technique and show that the regularization term added for privacy can be used to attain the optimal excess population loss by exploiting the stability-inducing property of regularization. The objective perturbation algorithm $\mathcal{A}_{\mathsf{ObjP}}$ is described in Algorithm 2.

In addition to smoothness and convexity, we make the following assumption on the loss.

**Assumption 5.1.** *For all $z \in \mathcal{Z}$, $\ell(\cdot, z)$ is twice-differentiable, and the rank of its Hessian $\nabla^2 \ell(\mathbf{w}, z)$ at any $\mathbf{w} \in \mathcal{W}$ is at most 1.*

---

**Algorithm 2** $\mathcal{A}_{\mathsf{ObjP}}$: Objective Perturbation for convex, smooth losses

---

**Input:** Private dataset: $S = (z_1, \ldots, z_n) \in \mathcal{Z}^n$, $L$-Lipschitz, $\beta$-smooth, convex loss function $\ell$, convex set $\mathcal{W} \subseteq \mathbb{R}^d$, privacy parameters $\epsilon \leq 1$, $\delta \leq 1/n^2$, regularization parameter $\lambda$.
1: Sample $\mathbf{G} \sim \mathcal{N}\left(\mathbf{0}, \sigma^2 \, \mathbb{I}_d\right)$, where $\sigma^2 = \frac{10 \, L^2 \, \log(1/\delta)}{\epsilon^2}$
2: **return** $\widehat{\mathbf{w}} = \arg\min_{\mathbf{w} \in \mathcal{W}} \widehat{\mathcal{L}}\left(\mathbf{w}; \, S\right) + \frac{\langle \mathbf{G}, \, \mathbf{w} \rangle}{n} + \lambda \|\mathbf{w}\|^2$, where $\widehat{\mathcal{L}}(\mathbf{w}; \, S) \triangleq \frac{1}{n} \sum_{i=1}^{n} \ell(\mathbf{w}, \, z_i)$.

---

**Note:** Unlike in [KST12], the regularization term as appears in $\mathcal{A}_{\mathsf{ObjP}}$ is not normalized by $n$. Hence, whenever the results from [KST12] are used here, the regularization parameter in their statements should be replaced with $n\lambda$. This presentation choice is more consistent with literature on regularization.

The privacy guarantee of $\mathcal{A}_{\mathsf{ObjP}}$ follows directly from [KST12]:

**Theorem 5.2** (Privacy guarantee of $\mathcal{A}_{\mathsf{ObjP}}$, restatement of Theorem 2 in [KST12])**.** *Suppose that Assumption 5.1 holds and that the smoothness parameter satisfies $\beta \leq \epsilon \, n \, \lambda$. Then, $\mathcal{A}_{\mathsf{ObjP}}$ is $(\epsilon, \delta)$-differentially private.*

We now state our main result for this section showing that, with appropriate setting for $\lambda$, $\mathcal{A}_{\mathsf{ObjP}}$ yields the optimal population loss.

**Theorem 5.3** (Excess population loss of $\mathcal{A}_{\mathsf{ObjP}}$)**.** *Let $\mathcal{D}$ be any distribution over $\mathcal{Z}$, and let $S \sim \mathcal{D}^n$. Suppose that Assumption 5.1 holds. Suppose that $\mathcal{W}$ is $M$-bounded. In $\mathcal{A}_{\mathsf{ObjP}}$, set $\lambda = \frac{2 \, L}{M} \sqrt{\frac{2}{n} + \frac{4 \, d \, \log(1/\delta)}{\epsilon^2 \, n^2}}$. Then, we have*

$$\Delta\mathcal{L}\left(\mathcal{A}_{\mathsf{ObjP}}; \, \mathcal{D}\right) \leq 2 \, M \, L \, \sqrt{\frac{2}{n} + \frac{4 \, d \, \log(1/\delta)}{\epsilon^2 \, n^2}} = O\left(M \, L \cdot \max\left(\frac{1}{\sqrt{n}}, \, \frac{\sqrt{d \, \log(1/\delta)}}{\epsilon \, n}\right)\right).$$

**Note:** According to Theorem 5.2, the privacy of $\mathcal{A}_{\mathsf{ObjP}}$ entails the assumption that $\beta \le \epsilon\, n\, \lambda$. With the setting of $\lambda$ in Theorem 5.3, it would suffice to assume that $\beta \le \frac{2\,\epsilon\,L}{M}\sqrt{2\,n + 4\,d\,\log(1/\delta)}$.

To prove the above theorem, we use the following lemmas.

**Lemma 5.4** (Excess empirical loss of $\mathcal{A}_{\mathsf{ObjP}}$, restatement of Theorem 26 in [KST12]). *Let $S \sim \mathcal{Z}^n$. Under Assumption 5.1, the excess empirical loss of $\mathcal{A}_{\mathsf{ObjP}}$ satisfies*

$$\mathbb{E}\left[\widehat{\mathcal{L}}(\widehat{\mathbf{w}}; S)\right] - \min_{\mathbf{w}\in\mathcal{W}} \widehat{\mathcal{L}}(\mathbf{w}; S) \le \frac{16\,L^2\,d\,\log(1/\delta)}{n^2\,\epsilon^2\,\lambda} + \lambda\,M^2.$$

*where the expectation is taken over the Gaussian noise in $\mathcal{A}_{\mathsf{ObjP}}$.*

**Lemma 5.5** ([SSBD14]). *Let $f : \mathcal{W} \times \mathcal{Z} \to \mathbb{R}$ be a convex, $\rho$-Lipschitz loss, and let $\lambda > 0$. Let $S = (z_1, \ldots, z_n) \sim \mathcal{Z}^n$. Let $\mathcal{A}$ be an algorithm that outputs $\widetilde{\mathbf{w}} = \arg\min_{\mathbf{w}\in\mathcal{W}}\left(\widehat{\mathcal{F}}(\mathbf{w};\ S) + \lambda\,\|\mathbf{w}\|^2\right)$, where $\widehat{\mathcal{F}}(\mathbf{w};\ S) = \frac{1}{n}\sum_{i=1}^{n} f(\mathbf{w},\ z_i)$. Then, $\mathcal{A}$ is $\frac{2\,\rho^2}{\lambda\,n}$-uniformly stable.*

**Proof of Theorem 5.3**

Fix any realization of the noise vector $\mathbf{G}$. For every $\mathbf{w}, z$ define $f_{\mathbf{G}}(\mathbf{w}, z) \triangleq \ell(\mathbf{w},\ z) + \frac{\langle\mathbf{G},\mathbf{w}\rangle}{n}$. Note that $f_{\mathbf{G}}$ is $\left(L + \frac{\|\mathbf{G}\|}{n}\right)$-Lipschitz. For any $S = (z_1, \ldots, z_n) \in \mathcal{Z}^n$, let $\widehat{\mathcal{F}}_{\mathbf{G}}(\mathbf{w}; S) \triangleq \frac{1}{n}\sum_{i=1}^{n} f_{\mathbf{G}}(\mathbf{w}, z_i)$. Hence, the output of $\mathcal{A}_{\mathsf{ObjP}}$ can be written as $\widehat{\mathbf{w}} = \arg\min_{\mathbf{w}\in\mathcal{W}} \widehat{\mathcal{F}}_{\mathbf{G}}(\mathbf{w};\ S) + \lambda\,\|\mathbf{w}\|^2$. Define $\mathcal{F}_{\mathbf{G}}(\mathbf{w};\ \mathcal{D}) \triangleq \mathbb{E}_{z\sim\mathcal{D}}[f_{\mathbf{G}}(\mathbf{w},\ z)]$. Thus, by combining Lemmas 5.5 and 2.2, we have $\mathbb{E}_{S\sim\mathcal{D}^n}\left[\mathcal{F}_{\mathbf{G}}(\widehat{\mathbf{w}};\ \mathcal{D}) - \widehat{\mathcal{F}}_{\mathbf{G}}(\widehat{\mathbf{w}};\ S)\right] \le \frac{2\left(L + \frac{\|\mathbf{G}\|}{n}\right)^2}{\lambda\,n}$. On the other hand, note that $\mathcal{F}_{\mathbf{G}}(\widehat{\mathbf{w}};\ \mathcal{D}) - \widehat{\mathcal{F}}_{\mathbf{G}}(\widehat{\mathbf{w}};\ S) = \mathcal{L}(\widehat{\mathbf{w}};\ \mathcal{D}) - \widehat{\mathcal{L}}(\widehat{\mathbf{w}};\ S)$ since the linear term cancels out. Hence,

$$\mathbb{E}_{S\sim\mathcal{D}^n}\left[\mathcal{L}(\widehat{\mathbf{w}};\ \mathcal{D}) - \widehat{\mathcal{L}}(\widehat{\mathbf{w}};\ S)\right] \le 2\,\frac{\left(L + \frac{\|\mathbf{G}\|}{n}\right)^2}{\lambda\,n}$$

By taking expectation over $\mathbf{G} \sim \mathcal{N}\left(\mathbf{0}, \sigma^2\mathbb{I}_d\right)$ as well, we get $\mathbb{E}\left[\mathcal{L}(\widehat{\mathbf{w}};\ \mathcal{D}) - \widehat{\mathcal{L}}(\widehat{\mathbf{w}};\ S)\right] \le 8\frac{L^2}{\lambda\,n}$.

Now, observe that:

$$\Delta\mathcal{L}\left(\mathcal{A}_{\mathsf{ObjP}};\mathcal{D}\right) \le \mathbb{E}\left[\widehat{\mathcal{L}}(\widehat{\mathbf{w}};\ S) - \min_{\mathbf{w}\in\mathcal{W}}\widehat{\mathcal{L}}(\mathbf{w};\ S)\right] + \mathbb{E}\left[\mathcal{L}(\widehat{\mathbf{w}};\ \mathcal{D}) - \widehat{\mathcal{L}}(\widehat{\mathbf{w}};\ S)\right]$$

$$\le \frac{8}{\lambda}\left(\frac{2\,L^2\,d\,\log(1/\delta)}{\epsilon^2\,n^2} + \frac{L^2}{n}\right) + \lambda\,M^2$$

where we use Lemma 5.4 in the last bound. Optimizing this bound in $\lambda$ yields the result.

*A note on the rank assumption:* The assumption on the rank of $\bigtriangledown^2\ell(\mathbf{w}, z)$ can actually be relaxed (using similar argument in [INS+19]) to a rank of $\widetilde{O}\left(\frac{L\sqrt{n+d}}{\beta M}\right)$ without affecting the asymptotic population loss guarantees (see the full version [BFTT19] for a discussion.)

**Efficient Objective Perturbation:** The privacy guarantee of the standard objective perturbation technique is given only when the output is the exact minimizer [CMS11, KST12]. Exact minimization is not usually attainable in practice. We give a practical version of algorithm $\mathcal{A}_{\mathsf{ObjP}}$ that attains the same guarantees of privacy and optimal population loss as $\mathcal{A}_{\mathsf{ObjP}}$, and in addition, makes only $O(n\log n)$ number of gradient evaluations. The main idea is to first obtain an approximate minimizer $\tilde{\mathbf{w}}$ that is sufficiently close to the true minimizer, and then perturb $\tilde{\mathbf{w}}$ with only a small amount of Gaussian noise to ensure privacy. The extra error due to the little noise added in the last step ends up having a trivial impact on the population loss. Hence, the algorithm achieves the same guarantees as $\mathcal{A}_{\mathsf{ObjP}}$. Crucially, it attains the optimal population loss in an efficient manner. In particular, we use Stochastic Variance Reduced Gradient Descent (SVRG) [JZ13, XZ14] to perform the optimization step, which leads to a construction with $O(n\log n)$ number of gradient evaluations. Detailed discussion can be found in the full version [BFTT19].

**Acknowledgements**

We thank Adam Smith, Thomas Steinke and Jon Ullman for the insightful discussions of the problem at the early stages of this project. We are also grateful to Tomer Koren for bringing the Moreau-Yosida smoothing technique to our attention. R. Bassily's research is supported by NSF Awards AF-1908281, SHF-1907715, Google Faculty Research Award, and OSU faculty start-up support. A. Thakurta's research is supported by NSF Awards TRIPODS+X-1839317, AF-1908281, TRIPODS-1740850, and Google Faculty Research Award.

## Footnotes

[2] In this Introduction, we are concerned with the dependence on $d$ and $n$, for $(\epsilon, \delta)$-DP. We suppress the dependence on $\delta$ and on parameters of the loss function such as Lipschitz constant and the constraint set radius.

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
