[Supplementary Material]

# Private Stochastic Convex Optimization with Optimal Rates

### Abstract

We study the differentially private analogue of the classical stochastic convex optimization problem, where we wish to approximately minimize the expected loss over a distribution of convex and Lipschitz loss functions, given access to samples from the the distribution. We present algorithms that achieve optimal rates and are computationally efficient. Our work builds on the long line of work on the empirical versions of this question, the Differentially Private Empirical Risk Minimization problem (DP-ERM). Previous attempts to derive population risk bounds have yielded suboptimal results and the correct rate was previously unknown. By drawing on the connection with algorithmic stability, we show that up to logarithmic factors, the optimal private population risk is equal to the larger of the optimal non-private population risk, and the optimal private empirical risk. This implies that, contrary to intuition based on private ERM, private SCO has asymptotically the same rate of $1/\sqrt{n}$ as non-private SCO in the parameter regime most common in practice.

## 1   Introduction

Many fundamental problems in machine learning reduce to the problem of minimizing the expected loss (a.k.a. population loss) $\mathcal{L}(\mathbf{w}) = \mathbb{E}_{z \sim \mathcal{D}}[\ell(\mathbf{w}, z)]$ for convex loss functions of $\mathbf{w}$ given access to samples $z_1, \ldots, z_n$ from the data distribution $\mathcal{D}$. This problem arises in various settings, such as estimating the mean of a distribution, least squares regression, or minimizing a convex surrogate loss for a classification problem. This problem is commonly referred to as *Stochastic Convex Optimization* (SCO) and has been the subject of extensive study in machine learning and optimization [SSBD14]. In this work we study this problem with the additional constraint of differential privacy.

A closely related problem is that of minimizing the loss $\widehat{\mathcal{L}}(\mathbf{w}) = \frac{1}{n} \sum_i \ell(\mathbf{w}, z_i)$ on the sampled set of functions, often known as Empirical Risk Minimization (ERM). The problem of private ERM has been well-studied and tight upper and lower bounds are known for private ERM. At first glance these two problems may appear to be essentially the same as an optimal algorithm for minimizing the empirical risk should also achieve the best bounds for the population risk itself, i.e. the best approach to private SCO is to use the best private ERM.

This simple intuition is unfortunately false, even in the non-private case. A natural approach of bounding the population loss is by proving an upper bound on $\mathbb{E}_{z_1,\ldots,z_n} \left[ \sup_{\mathbf{w}} (\mathcal{L}(\mathbf{w}) - \widehat{\mathcal{L}}(\mathbf{w})) \right]$. This is known as *uniform convergence*. There are examples of distributions over losses where uniform convergence based bounds are provably sub-optimal. For example, for convex Lipschitz losses in $d$-dimensional Euclidean space, the best bound on the population loss achievable via uniform convergence is $\Omega(\sqrt{d/n})$ [Fel16]. In contrast, SGD is known to achieve excess loss of $O(\sqrt{\frac{1}{n}})$ which is independent of the dimension.

As a result, in the high-dimensional settings often considered in modern ML the case (when $n = \Theta(d)$), the optimal achievable excess loss is $O(\sqrt{\frac{1}{n}})$, whereas the uniform convergence bound is $\Omega(1)$.

This discrepancy implies that using private ERM and appealing to uniform convergence will not lead to optimal bounds for private SCO. The first work to address the population loss for private SCO is [BST14] which gives bounds based on several natural approaches. Their first approach is to use the generalization properties of differential privacy itself to bound the gap between the empirical and population losses [DFH$^+$15, BNS$^+$16], and thus derive bounds for SCO from bounds on ERM. This approach leads to a suboptimal bound for private SCO (specifically[1], $\approx \max\left(\frac{d^{\frac{1}{4}}}{\sqrt{n}}, \frac{\sqrt{d}}{\epsilon n}\right)$ [BST14, Sec. F]). For the important case of $d$ being on the order of $n$ and $\epsilon$ being on the order of one this results in $\Omega(n^{-\frac{1}{4}})$ bound on excess population loss. Their second approach uses stability induced by regularizing the empirical loss before it is minimized via a private ERM algorithm for strongly convex losses. This technique also yields a suboptimal bound on the excess population loss $\approx \frac{d^{\frac{1}{4}}}{\sqrt{\epsilon} n}$. Finally, they note that for some simple subclasses of convex functions such as Generalized Linear Models, uniform convergence bounds on the order of $1/\sqrt{n}$ are known [KST08] and therefore optimal bounds for private SCO are implied by optimal bounds on private ERM [JT14, BST14].

There are two natural lower bounds that apply to private SCO. Naturally, it cannot achieve better bounds than non-private SCO, that is the excess loss is $\Omega(\sqrt{\frac{1}{n}})$. Further it is not hard to show that lower bounds for private ERM translate to essentially the same lower bound for private SCO, leading to a lower bound of the form $\Omega(\frac{\sqrt{d}}{\epsilon n})$. We give complete argument for the lower bound in Appendix C .

In this work, we address the question:

*What is the optimal rate for the excess loss of private SCO? Is the rate of $O\left(\sqrt{\frac{1}{n}} + \frac{\sqrt{d}}{\epsilon n}\right)$ achievable?*

## 1.1 Our contribution

We show that the optimal rate of $O\left(\sqrt{\frac{1}{n}} + \frac{\sqrt{d}}{\epsilon n}\right)$ is achievable. In particular, we obtain the statistically optimal rate of $O(1/\sqrt{n})$ whenever $d = O(n)$. This is in contrast to the situation for private ERM where the cost of privacy grows with the dimension for all $n$.

In fact, under relatively mild smoothness assumptions, it is achieved by a variant of the standard noisy mini-batch SGD. The parameters of the scheme need to be tuned carefully to satisfy a delicate balance. The classical analyses for non-private SCO depend crucially on making only one pass over the dataset. However, a single pass noisy SGD is not sufficiently accurate as we need a non-trivial amount of noise in each step to carry out the privacy analysis. We rely instead on a different approach to generalization, known as *uniform stability* [BE02]. The stability parameter degrades with the number of passes over the dataset [HRS15, FV19], while the empirical accuracy improves as we make more passes. We show that $\approx \max(\sqrt{n}, \sqrt{d})$ passes are sufficient to get a good enough approximation to the ERM, while still ensuring sufficient uniform stability to bound the population objective. We also need to choose the batch size carefully since privacy amplification by subsampling technique used in prior works [BST14] does not cover the desired range of $\epsilon$ when few steps of SGD are taken.

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

# 3   Private SCO via Mini-batch Noisy SGD

In this section, we consider the setting where the loss $\ell$ is convex, Lipschitz, and smooth. We give a technique that is based on a mini-batch variant of Noisy Stochastic Gradient Descent (NSGD) algorithm [BST14, ACG$^+$16].

**Theorem 3.1** (Privacy guarantee of $\mathcal{A}_{\mathsf{NSGD}}$). *Algorithm 1 is $(\epsilon, \delta)$-differentially private.*

*Proof.* The proof follows from [ACG$^+$16, Theorem 1], which gives a tight privacy analysis for mini-batch NSGD via the Moments Accountant technique and privacy amplification via sampling. We note that the setting of the mini-batch size in Step 2 of Algorithm 1 satisfies the condition in [ACG$^+$16, Theorem 1] (we obtain here an explicit value for the universal constants in the aforementioned theorem in that reference). We also note that the setting of the Gaussian noise in [ACG$^+$16] is not normalized by the mini-batch size, and hence the noise variance reported in [ACG$^+$16, Theorem 1] is larger than our setting of $\sigma^2$ by a factor of $m^2$. $\qquad\square$

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

The convexity and $\beta$-smoothness together with properties 2 and 3 are fairly standard and the proof can be found in the aforementioned references. The fact that $f_\beta$ is $2L$-Lipschitz follows easily from property 3. We include the proof of this fact in Appendix B for completeness.

Let $\ell : \mathcal{W} \times \mathcal{Z} \to \mathbb{R}$ be a convex, $L$-Lipschitz loss. For any $z \in \mathcal{Z}$, let $\ell_\beta(\cdot, z)$ denote the $\beta$-Moreau envelope of $\ell(\cdot, z)$. For a dataset $S = (z_1, \ldots, z_n) \in \mathcal{Z}^n$, let $\widehat{\mathcal{L}}_\beta(\cdot; S) \triangleq \frac{1}{n}\sum_{i=1}^n \ell_\beta(\cdot, z_i)$ be the empirical risk w.r.t. the $\beta$-smoothed loss. For any distribution $\mathcal{D}$, let $\mathcal{L}_\beta(\cdot; \mathcal{D}) \triangleq \mathop{\mathbb{E}}_{z \sim \mathcal{D}} [\ell(\cdot, z)]$ denote the corresponding population loss. The following theorem asserts that, with an appropriate setting for $\beta$, running $\mathcal{A}_{\mathsf{NSGD}}$ over the $\beta$-smoothed losses $\ell_\beta(\cdot, z_i)$, $i \in [n]$ yields the optimal population loss w.r.t. the original non-smooth loss $\ell$.

**Theorem 4.4** (Excess population loss for non-smooth losses via smoothing). *Let $\mathcal{D}$ be any distribution over $\mathcal{Z}$. Let $S = (z_1, \ldots, z_n) \sim \mathcal{D}^n$. Let $\beta = \frac{L}{M} \cdot \min\left( \frac{\sqrt{n}}{4}, \frac{\epsilon n}{8\sqrt{d \log(1/\delta)}} \right)$. Suppose we run $\mathcal{A}_{\mathsf{NSGD}}$ (Algorithm 1) over the $\beta$-smoothed version of $\ell$ associated with the points in $S$: $\{\ell_\beta(\cdot, z_i), i \in [n]\}$. Let $\eta$ and $T$ be set as in Theorem 3.2. Then, the excess population loss of the output of $\mathcal{A}_{\mathsf{NSGD}}$ w.r.t. $\ell$ satisfies*

$$\Delta\mathcal{L}\left(\mathcal{A}_{\mathsf{NSGD}}; \mathcal{D}\right) \le 24\, M\, L \cdot \max\left( \frac{\sqrt{d \log(1/\delta)}}{\epsilon\, n}, \frac{1}{\sqrt{n}} \right)$$

*Proof.* Let $\overline{\mathbf{w}}_T$ be the output of $\mathcal{A}_{\mathsf{NSGD}}$. Using property 1 of Lemma 4.3 together with Theorem 3.2, we have

$$\mathop{\mathbb{E}}_{S \sim \mathcal{D}^n, \mathcal{A}_{\mathsf{NSGD}}} [\mathcal{L}_\beta(\overline{\mathbf{w}}_T; \mathcal{D})] - \min_{\mathbf{w} \in \mathcal{W}} \mathcal{L}_\beta(\mathbf{w}; \mathcal{D}) \le 20\, M\, L \cdot \max\left( \frac{\sqrt{d \log(1/\delta)}}{\epsilon\, n}, \frac{1}{\sqrt{n}} \right).$$

Now, by property 2 of Lemma 2 and the setting of $\beta$ in the theorem statement, for every $\mathbf{w} \in \mathcal{W}$, we have

$$\mathcal{L}_\beta(\mathbf{w}; \mathcal{D}) \le \mathcal{L}(\mathbf{w}; \mathcal{D}) \le \mathcal{L}_\beta(\mathbf{w}; \mathcal{D}) + 2\, M\, L \cdot \max\left( \frac{1}{\sqrt{n}}, \frac{2\sqrt{d \log(1/\delta)}}{\epsilon\, n} \right).$$

Putting these together gives the stated result. $\qquad\square$

## Computationally efficient algorithm $\mathcal{A}_{\mathsf{ProxGD}}$ (NSGD + Prox)

Computing the Moreau envelope of a function is computationally inefficient in general. However, by property 3 of Lemma 4.3, we note that evaluating the gradient of Moreau envelope at any point can be attained by evaluating the proximal operator of the function at that point. Evaluating the proximal operator is equivalent to minimizing a strongly convex function (see Definition 4.2). This can be approximated efficiently, e.g., via gradient descent. Since our $\mathcal{A}_{\mathsf{NSGD}}$ algorithm (Algorithm 1) requires only sufficiently accurate gradient evaluations, we can hence use an efficient, approximate proximal operator to approximate the gradient of the smoothed losses. The gradient evaluations in $\mathcal{A}_{\mathsf{NSGD}}$ will thus be replaced with such approximate gradients evaluated via the approximate proximal operator. The resulting algorithm, referred to as $\mathcal{A}_{\mathsf{ProxGD}}$, will approximately minimize the smoothed empirical loss without actually computing the smoothed losses.

**Definition 4.5** (Approximate prox operator). *We say that $\widehat{\mathrm{prox}}_f$ is an $\xi$-approximate proximal operator of $\mathrm{prox}_f$ for a function $f : \mathcal{W} \to \mathbb{R}$ if $\forall \mathbf{w} \in \mathcal{W}$, $\|\widehat{\mathrm{prox}}_f(\mathbf{w}) - \mathrm{prox}_f(\mathbf{w})\| \le \xi$.*

**Fact 4.6.** *Let $\mathcal{W} \subset \mathbb{R}^d$ be $M$-bounded. Let $f : \mathcal{W} \to \mathbb{R}$ be convex, $L$-Lipschitz function. Suppose $\beta \ge \frac{L}{M}$. For all $\xi > 0$, there is $\xi$-approximate $\widehat{\mathrm{prox}}_{f/\beta}$ such that for each $\mathbf{w} \in \mathcal{W}$, computing $\widehat{\mathrm{prox}}_{f/\beta}(\mathbf{w})$ requires time that is equivalent to at most $\lceil \frac{8\,M^2}{\xi^2} \rceil$ gradient evaluations.*

This fact follows from the fact that $\mathrm{prox}_{f/\beta}(\mathbf{w}) = \arg\min_{\mathbf{v} \in \mathcal{W}} g_{\mathbf{w}}(\mathbf{v})$, where $g_{\mathbf{w}}(\mathbf{v}) \triangleq \frac{1}{\beta} f(\mathbf{v}) + \frac{1}{2}\|\mathbf{v} - \mathbf{w}\|^2$. This is minimization of $1$-strongly convex and $2\,M$-Lipschitz function over $\mathcal{W}$, The Lipschitz constant follows from the fact that $\beta \ge L/M$. Hence, one can run ordinary Gradient Descent to obtain an approximate minimizer. From a standard result on convergence of GD for strongly convex and Lipschitz functions [Bub15], in $\tau$ gradient steps we obtain an approximate $\mathbf{v}_\tau$ satisfying $g_{\mathbf{w}}(\mathbf{v}_\tau) - g_{\mathbf{w}}(\mathbf{v}^*) \le \frac{8\,M^2}{\tau}$, where $\mathbf{v}^* = \arg\min_{\mathbf{v} \in \mathcal{W}} g_{\mathbf{w}}(\mathbf{v})$. Since $g_{\mathbf{w}}$ is $1$-strongly convex, we get $\|\mathbf{v}_\tau - \mathbf{v}^*\| \le \sqrt{\frac{8\,M^2}{\tau}}$.

**Description of $\mathcal{A}_{\mathsf{ProxGD}}$:** The algorithm description follows exactly the same lines as $\mathcal{A}_{\mathsf{NSGD}}$ except that: (i) the input loss $\ell$ is now non-smooth, and (ii) for each iteration $t$, the gradient evaluation $\nabla\ell(\mathbf{w}_t, z)$ for each data point $z$ in the mini-batch is replaced with the evaluation of an approximate gradient of the smoothed loss $\ell_\beta(\cdot, z)$. The approximate gradient, denoted as $\widehat{\nabla}\ell_\beta(\mathbf{w}_t, z)$, is computed using an approximate proximal operator. Namely,

$$\widehat{\nabla}\ell_\beta(\mathbf{w}_t, z) := \beta \cdot \left( \mathbf{w}_t - \widehat{\mathrm{prox}}_{\ell_z/\beta}(\mathbf{w}_t) \right),$$

where $\ell_z \triangleq \ell(\cdot,\ z)$. Here, we use a computationally efficient $\xi$-approximate $\widehat{\mathrm{prox}}_{\ell_z/\beta}$ like the one in Fact 4.6 with $\xi$ set as

$$\xi := 4\,\frac{M}{n} \cdot \max\left( \frac{2\,\sqrt{d\,\log(1/\delta)}}{\epsilon\,n}, \frac{1}{\sqrt{n}} \right).$$

Note that the approximation error in the gradient $\|\widehat{\nabla}\ell_\beta(\mathbf{w}_t, z) - \nabla\ell_\beta(\mathbf{w}_t, z)\| \le \beta \cdot \xi$, and that $\beta \cdot \xi = \frac{L}{n}$, where $L$ is the Lipschitz constant of $\ell$.

**Running time of $\mathcal{A}_{\mathsf{ProxGD}}$:** if we use the approximate proximal operator in Fact 4.6, then it is easy to see that $\mathcal{A}_{\mathsf{ProxGD}}$ requires a number of gradient evaluations that is a factor of $n^2\,T$ more than $\mathcal{A}_{\mathsf{NSGD}}$, where $T = O\left( \max\left( n, \frac{\epsilon^2\,n^2}{d\,\log(1/\delta)} \right) \right)$. That is, the total number of gradient evaluations is $n^2 \cdot T^2 \cdot m$, where $m = O\left( \max\left( \sqrt{\epsilon\,n}, \sqrt{\frac{d\,\log(1/\delta)}{\epsilon}} \right) \right)$ is the mini-batch size.

We now argue that privacy, stability, and accuracy of the algorithm are preserved under the approximate proximal operator.

**Privacy:** Note that to bound the sensitivity of the approximate gradient of the mini-batch, it suffices to bound the norm of the approximate gradient. From the discussion above, note that $\forall\, z, \forall\, \mathbf{w} \in \mathcal{W}$, we have $\|\widehat{\nabla}\ell_\beta(\mathbf{w}, z)\| \le \|\widehat{\nabla}\ell_\beta(\mathbf{w}, z) - \nabla\ell_\beta(\mathbf{w}, z)\| + \|\nabla\ell_\beta(\mathbf{w}, z)\| \le L\,(1 + \frac{1}{n})$. Thus, the sensitivity remains basically the same as in the case where the algorithm is run with the exact gradients. Hence, the same privacy guarantee holds as in $\mathcal{A}_{\mathsf{NSGD}}$.

**Empirical error:** Note that the approximation error in the gradient of the mini-batch (due to the approximate proximal operation) can be viewed as a *fixed* error term of magnitude at most $\frac{L}{n}$ that is added to the Gaussian noise. Thus, the variance of the total noise (Gaussian noise + approximation error) in each iteration is at most $\widehat{\sigma}^2 = O\left( \frac{T\,L^2\,d\,\log(1/\delta)}{n^2\,\epsilon^2} \right) + \frac{L^2}{n^2} = O\left( \frac{T\,L^2\,d\,\log(1/\delta)}{n^2\,\epsilon^2} \right)$. Hence, we get the same bound on the excess empirical risk as in Lemma 3.3.

**Uniform stability:** This easily follows from the following facts. First, note that the additional approximation error due to gradient approximation is $\frac{L}{n}$. Second, the gradient update w.r.t. the exact gradient of the smoothed loss is non-expansive operation (which is the key fact in proving uniform stability of (stochastic) gradient methods [HRS15, FV19]), and hence the approximation error in the gradient is not going to be amplified by the gradient update step. Hence, for any trajectory of $T$ approximate gradient updates, the accumulated approximation error in the final output $\overline{\mathbf{w}}_T$ cannot exceed $\frac{T\eta L}{n}$. This cannot increase the final uniform stability bound by more than an additive term of $\frac{T\eta L^2}{n}$. Thus, we obtain basically the same bound in Lemma 3.4.

Putting these together, we have argued that $\mathcal{A}_{\mathsf{ProxGD}}$ is computationally efficient algorithm that achieves the optimal population loss bound in Theorem 4.4.

# 5    Private SCO via Objective Perturbation

In this section, we show that the technique known as objective perturbation [CMS11, KST12] can be used to attain optimal *population* loss for a large subclass of convex, smooth losses. In objective perturbation, the empirical loss is first perturbed by adding two terms: a *noisy* linear term and a regularization term. As shown in [CMS11, KST12], under some additional assumptions on the hessian of the loss, an appropriate random perturbation ensures differential privacy. The excess *empirical* loss of this technique for smooth convex losses was originally analyzed in the aforementioned works, and was shown to be optimal by the lower bound in [BST14]. We revisit this technique and show that the regularization term added for privacy can be used to attain the optimal excess population loss by exploiting the stability-inducing property of regularization.

In addition to smoothness and convexity of $\ell$, as in [CMS11, KST12], we also make the following assumption on the loss function.

**Assumption 5.1.** *For all $z \in \mathcal{Z}$, $\ell(\cdot,\ z)$ is twice-differentiable, and the rank of its Hessian $\nabla^2 \ell(\mathbf{w}, z)$ at any $\mathbf{w} \in \mathcal{W}$ is at most 1.*

The description of the objective perturbation algorithm $\mathcal{A}_{\mathsf{ObjP}}$ is given in Algorithm 2. The outline of the algorithm is the same as the one in [KST12] for the case of $(\epsilon, \delta)$-differential privacy.

---

**Algorithm 2** $\mathcal{A}_{\mathsf{ObjP}}$: Objective Perturbation for convex, smooth losses

---

**Input:** Private dataset: $S = (z_1, \ldots, z_n) \in \mathcal{Z}^n$, $L$-Lipschitz, $\beta$-smooth, convex loss function $\ell$, convex set $\mathcal{W} \subseteq \mathbb{R}^d$, privacy parameters $\epsilon \leq 1$, $\delta \leq 1/n^2$, regularization parameter $\lambda$.

1: Sample $\mathbf{G} \sim \mathcal{N}\left(\mathbf{0}, \sigma^2 \,\mathbb{I}_d\right)$, where $\sigma^2 = \frac{10\,L^2 \,\log(1/\delta)}{\epsilon^2}$

2: **return** $\widehat{\mathbf{w}} = \arg\min_{\mathbf{w} \in \mathcal{W}} \widehat{\mathcal{L}}\left(\mathbf{w};\ S\right) + \frac{\langle \mathbf{G}, \mathbf{w}\rangle}{n} + \lambda\|\mathbf{w}\|^2$, where $\widehat{\mathcal{L}}(\mathbf{w};\ S) \triangleq \frac{1}{n}\sum_{i=1}^{n} \ell(\mathbf{w},\ z_i)$.

---

**Note:** The regularization term as appears in $\mathcal{A}_{\mathsf{ObjP}}$ is of different scaling than the one that appears in [KST12]. In particular, the regularization term in [KST12] is normalized by $n$, whereas here it is not. Hence, whenever the results from [KST12] are used here, the regularization parameter in their statements should be replaced with $n\lambda$. This presentation choice is more consistent with literature on regularization.

The privacy guarantee of $\mathcal{A}_{\mathsf{ObjP}}$ is given in the following theorem, which follows directly from [KST12].

**Theorem 5.2** (Privacy guarantee of $\mathcal{A}_{\mathsf{ObjP}}$, restatement of Theorem 2 in [KST12])**.** *Suppose that Assumption 5.1 holds and that the smoothness parameter satisfies $\beta \leq \epsilon\,n\,\lambda$. Then, $\mathcal{A}_{\mathsf{ObjP}}$ is $(\epsilon, \delta)$-differentially private.*

We now state our main result for this section showing that, with appropriate setting for $\lambda$, $\mathcal{A}_{\mathsf{ObjP}}$ yields the optimal population loss.

**Theorem 5.3** (Excess population loss of $\mathcal{A}_{\mathsf{ObjP}}$). *Let $\mathcal{D}$ be any distribution over $\mathcal{Z}$, and let $S \sim \mathcal{D}^n$. Suppose that Assumption 5.1 holds. Suppose that $\mathcal{W}$ is $M$-bounded. In $\mathcal{A}_{\mathsf{ObjP}}$, set $\lambda = \frac{2L}{M}\sqrt{\frac{2}{n} + \frac{4\,d\,\log(1/\delta)}{\epsilon^2\,n^2}}$. Then, we have*

$$\Delta\mathcal{L}\left(\mathcal{A}_{\mathsf{ObjP}};\ \mathcal{D}\right) \leq 2\,M\,L\,\sqrt{\frac{2}{n} + \frac{4\,d\,\log(1/\delta)}{\epsilon^2\,n^2}} = O\left(M\,L\cdot\max\left(\frac{1}{\sqrt{n}},\ \frac{\sqrt{d\,\log(1/\delta)}}{\epsilon\,n}\right)\right).$$

**Note:** According to Theorem 5.2, $(\epsilon,\delta)$-differential privacy of $\mathcal{A}_{\mathsf{ObjP}}$ entails the assumption that $\beta \leq \epsilon\,n\,\lambda$. With the setting of $\lambda$ in Theorem 5.3, it would suffice to assume that $\beta \leq \frac{2\,\epsilon\,L}{M}\sqrt{2\,n + 4\,d\,\log(1/\delta)}$.

To prove the above theorem, we use the following lemmas.

**Lemma 5.4** (Excess empirical loss of $\mathcal{A}_{\mathsf{ObjP}}$, restatement of Theorem 26 in [KST12]). *Let $S \sim \mathcal{Z}^n$. Under Assumption 5.1, the excess empirical loss of $\mathcal{A}_{\mathsf{ObjP}}$ satisfies*

$$\mathbb{E}\left[\widehat{\mathcal{L}}(\widehat{\mathbf{w}}; S)\right] - \min_{\mathbf{w}\in\mathcal{W}}\widehat{\mathcal{L}}(\mathbf{w}; S) \leq \frac{16\,L^2\,d\,\log(1/\delta)}{n^2\,\epsilon^2\,\lambda} + \lambda\,M^2.$$

*where the expectation is taken over the Gaussian noise in $\mathcal{A}_{\mathsf{ObjP}}$.*

The next lemma states a well-known fact, namely, regularized empirical risk minimization is uniformly stable.

**Lemma 5.5** ([SSSSS09, SSBD14]). *Let $f : \mathcal{W} \times \mathcal{Z} \to \mathbb{R}$ be a convex, $\rho$-Lipschitz loss, and let $\lambda > 0$. Let $S = (z_1,\ldots,z_n) \sim \mathcal{Z}^n$. Let $\mathcal{A}$ be an algorithm that outputs $\widetilde{\mathbf{w}} = \arg\min_{\mathbf{w}\in\mathcal{W}}\left(\widehat{\mathcal{F}}(\mathbf{w};\ S) + \lambda\|\mathbf{w}\|^2\right)$, where $\widehat{\mathcal{F}}(\mathbf{w};\ S) = \frac{1}{n}\sum_{i=1}^n f(\mathbf{w},\ z_i)$. Then, $\mathcal{A}$ is $\frac{2\,\rho^2}{\lambda\,n}$-uniformly stable.*

**Proof of Theorem 5.3**

Fix any realization of the noise vector $\mathbf{G}$. For every $\mathbf{w} \in \mathcal{W}, z \in \mathcal{Z}$, define $f_{\mathbf{G}}(\mathbf{w},z) \triangleq \ell(\mathbf{w},\ z) + \frac{\langle\mathbf{G},\mathbf{w}\rangle}{n}$. Note that $f_{\mathbf{G}}$ is $\left(L + \frac{\|\mathbf{G}\|}{n}\right)$-Lipschitz. For any dataset $S = (z_1,\ldots,z_n) \in \mathcal{Z}^n$, define $\widehat{\mathcal{F}}_{\mathbf{G}}(\mathbf{w};S) \triangleq \frac{1}{n}\sum_{i=1}^n f_{\mathbf{G}}(\mathbf{w},z_i)$. Hence, the output $\widehat{\mathbf{w}}$ of $\mathcal{A}_{\mathsf{ObjP}}$ on input dataset $S$ can be written as $\widehat{\mathbf{w}} = \arg\min_{\mathbf{w}\in\mathcal{W}}\widehat{\mathcal{F}}_{\mathbf{G}}(\mathbf{w};\ S) + \lambda\|\mathbf{w}\|^2$. Define $\mathcal{F}_{\mathbf{G}}(\mathbf{w};\ \mathcal{D}) \triangleq \mathbb{E}_{z\sim\mathcal{D}}[f_{\mathbf{G}}(\mathbf{w},\ z)]$. Thus, for any fixed $\mathbf{G}$, by combining Lemma 5.5 with Lemma 2.2, we have $\mathbb{E}_{S\sim\mathcal{D}^n}\left[\mathcal{F}_{\mathbf{G}}(\widehat{\mathbf{w}};\ \mathcal{D}) - \widehat{\mathcal{F}}_{\mathbf{G}}(\widehat{\mathbf{w}};\ S)\right] \leq \frac{2\left(L + \frac{\|\mathbf{G}\|}{n}\right)^2}{\lambda\,n}$. On the other hand, note that for any dataset $S$, we always have $\mathcal{F}_{\mathbf{G}}(\widehat{\mathbf{w}};\ \mathcal{D}) - \widehat{\mathcal{F}}_{\mathbf{G}}(\widehat{\mathbf{w}};\ S) = \mathcal{L}(\widehat{\mathbf{w}};\ \mathcal{D}) - \widehat{\mathcal{L}}(\widehat{\mathbf{w}};\ S)$ since the linear term cancels out. Hence, the expected generalization error (w.r.t. $S$) satisfies

$$\mathbb{E}_{S\sim\mathcal{D}^n}\left[\mathcal{L}(\widehat{\mathbf{w}};\ \mathcal{D}) - \widehat{\mathcal{L}}(\widehat{\mathbf{w}};\ S)\right] \leq 2\,\frac{\left(L + \frac{\|\mathbf{G}\|}{n}\right)^2}{\lambda\,n}$$

Now, by taking expectation over $\mathbf{G} \sim \mathcal{N}\left(\mathbf{0}, \sigma^2\mathbb{I}_d\right)$ as well, we arrive at

$$\mathbb{E}\left[\mathcal{L}(\widehat{\mathbf{w}};\ \mathcal{D}) - \widehat{\mathcal{L}}(\widehat{\mathbf{w}};\ S)\right] \leq 2\,L^2\,\frac{\left(1 + \frac{\sqrt{10\,d\,\log(1/\delta)}}{\epsilon\,n}\right)^2}{\lambda\,n} \leq 8\,\frac{L^2}{\lambda\,n} \tag{2}$$

where we assume $\frac{\sqrt{10\, d\, \log(1/\delta)}}{\epsilon\, n} \leq 1$ (since otherwise we would have the trivial error).

Now, observe that:

$$\Delta\mathcal{L}\left(\mathcal{A}_{\mathsf{ObjP}}; \mathcal{D}\right) = \mathbb{E}\left[\mathcal{L}(\widehat{\mathbf{w}}; \mathcal{D})\right] - \min_{\mathbf{w}\in\mathcal{W}} \mathcal{L}(\mathbf{w};\ \mathcal{D})$$

$$\leq \mathbb{E}\left[\widehat{\mathcal{L}}(\widehat{\mathbf{w}};\ S) - \min_{\mathbf{w}\in\mathcal{W}} \widehat{\mathcal{L}}(\mathbf{w};\ S)\right] + \mathbb{E}\left[\mathcal{L}(\widehat{\mathbf{w}};\ \mathcal{D}) - \widehat{\mathcal{L}}(\widehat{\mathbf{w}};\ S)\right]$$

$$\leq \frac{8}{\lambda}\left(\frac{2\, L^2\, d\, \log(1/\delta)}{\epsilon^2\, n^2} + \frac{L^2}{n}\right) + \lambda\, M^2$$

where the second inequality follows from the fact that $\displaystyle\mathbb{E}_{S\sim\mathcal{D}^n}\left[\min_{\mathbf{w}\in\mathcal{W}} \widehat{\mathcal{L}}(\mathbf{w};\ S)\right] \leq \min_{\mathbf{w}\in\mathcal{W}} \mathbb{E}_{S\sim\mathcal{D}^n}\left[\widehat{\mathcal{L}}(\mathbf{w};\ S)\right] = \min_{\mathbf{w}\in\mathcal{W}} \mathcal{L}(\mathbf{w};\ \mathcal{D})$, and the last bound follows from combining (2) with Lemma 5.4. Optimizing this bound in $\lambda$ yields the setting of $\lambda$ in the theorem statement. Plugging that setting of $\lambda$ into the bound yield the stated bound on the excess population loss.

***A note on the rank assumption:*** While in this section we presented our result under the assumption that rank of $\bigtriangledown^2 \ell(\mathbf{w}, z)$ is at most one, one can extend the analysis (by using similar argument in [INS$^+$19]) to a rank of $\widetilde{O}\left(\frac{L\sqrt{n+d}}{\beta M}\right)$ without affecting the asymptotic population loss guarantees. In general to ensure differential privacy to $\mathcal{A}_{\mathsf{ObjP}}$, one only need the following assumption involving the Hessian of individual losses: $\left|\det\left(\mathbb{I} + \frac{\bigtriangledown^2 \ell(\mathbf{w},z)}{\lambda}\right)\right| \leq e^{\epsilon/2}$ for all $z \in \mathcal{Z}$ and $\mathbf{w} \in \mathcal{W}$, rather than a constraint on the rank. We defer this general analysis to the full version.

## 5.1 Oracle Efficient Objective Perturbation

The privacy guarantee of the standard objective perturbation technique is given only when the output is the exact minimizer [CMS11, KST12]. In practice, we usually cannot attain the exact minimizer, but rather obtain an approximate minimizer via efficient optimization methods. Hence, in this section we focus on providing a practical version of algorithm $\mathcal{A}_{\mathsf{ObjP}}$, called *approximate objective perturbation* (Algorithm $\mathcal{A}_{\mathsf{ObjP-App}}$), that i) is $(\epsilon, \delta)$-differentially private, ii) achieves the same optimal population loss as $\mathcal{A}_{\mathsf{ObjP}}$, and iii) only makes $O(n \log n)$ evaluations of the gradient $\bigtriangledown_{\mathbf{w}} \ell(\mathbf{w}, z)$ at any $\theta \in \mathcal{W}$ and $z \in \mathcal{Z}$. The main idea in $\mathcal{A}_{\mathsf{ObjP-App}}$ is to first obtain a $\mathbf{w}_2$ that ensures $\mathcal{J}(\mathbf{w}_2; S) - \min_{\mathcal{W}} \mathcal{J}(\mathbf{w}; S)$ is at most $\alpha$, and then perturb $\mathbf{w}_2$ with Gaussian noise to "fuzz" the difference between $\mathbf{w}_2$ and the true minimizer. In this work, we use Stochastic Variance Reduced Gradient Descent (SVRG) [JZ13, XZ14] as the optimization algorithm. This leads to a construction that requires near linear oracle complexity (i.e., number of gradient evaluations). In particular, $\mathcal{A}_{\mathsf{ObjP-App}}$ achieves oracle complexity of $O(n \log n)$ for optimal population loss.

**Theorem 5.6** (Privacy guarantee of $\mathcal{A}_{\mathsf{ObjP-App}}$). *Suppose that Assumption 5.1 holds and that the smoothness parameter satisfies $\beta \leq \epsilon\, n\, \lambda$. Then, Algorithm $\mathcal{A}_{\mathsf{ObjP-App}}$ is $(\epsilon, \delta)$-differentially private.*

*Proof.* Let $\mathbf{w}_1 = \arg\min_{\mathbf{w}\in\mathcal{W}} \underbrace{\widehat{\mathcal{L}}(\mathbf{w};\ S) + \frac{\langle \mathbf{G},\ \mathbf{w}\rangle}{n} + \lambda \|\mathbf{w}\|^2}_{\mathcal{J}(\mathbf{w},S)}$, and $\mathbf{w}_2 = \mathcal{O}(\mathcal{J}, \alpha)$, where $\mathcal{O}$ is the optimizer defined in Algorithm $\mathcal{A}_{\mathsf{ObjP-App}}$. Notice that one can compute $\mathbf{w}_2$ from the tuple $(\mathbf{w}_1, \mathbf{w}_2 - \mathbf{w}_1)$ by simple post-processing. Furthermore, the algorithm that outputs $\mathbf{w}_1$ is $(\epsilon/2, \delta/2)$-differentially private by Theorem 5.2. In the following, we will bound $\|\mathbf{w}_2 - \mathbf{w}_1\|$ in order to make $(\mathbf{w}_2 - \mathbf{w}_1)$ differentially private, conditioned on the knowledge of $\mathbf{w}_1$.

**Algorithm 3** $\mathcal{A}_{\mathsf{ObjP-App}}$: Approximate Objective Perturbation for convex, smooth losses

---

**Input:** Private dataset: $S = (z_1, \ldots, z_n) \in \mathcal{Z}^n$, $L$-Lipschitz, $\beta$-smooth, convex loss function $\ell$, convex set $\mathcal{W} \subseteq \mathbb{R}^d$, privacy parameters $\epsilon \leq 1$, $\delta \leq 1/n^2$, regularization parameter $\lambda$, Optimizer $\mathcal{O} : \mathcal{F} \times [0, 1] \to \mathcal{W}$ (where $\mathcal{F}$ is the class of objectives, and the other argument is the optimization accuracy), $\alpha \in [0, 1]$ : optimization accuracy.

1: Sample $\mathbf{G} \sim \mathcal{N}\left(\mathbf{0}, \sigma^2\, \mathbb{I}_d\right)$, where $\sigma^2 = \frac{20\, L^2\, \log(1/\delta)}{\epsilon^2}$.
2: Let $\mathcal{J}(\mathbf{w}; S) = \widehat{\mathcal{L}}(\mathbf{w};\, S) + \frac{\langle \mathbf{G},\, \mathbf{w} \rangle}{n} + \lambda \|\mathbf{w}\|^2$, where $\widehat{\mathcal{L}}(\mathbf{w};\, S) \triangleq \frac{1}{n}\sum_{i=1}^{n} \ell(\mathbf{w},\, z_i)$.
3: **return** $\widehat{\mathbf{w}} = \mathsf{Proj}_{\mathcal{W}}\left[\mathcal{O}\left(\mathcal{J}, \alpha\right) + \mathbf{H}\right]$, where $\mathbf{H} \sim \mathcal{N}\left(\mathbf{0}, \sigma^2\, \mathbb{I}_d\right)$, and $\sigma^2 = \frac{40\alpha\, \log(1/\delta)}{\lambda\epsilon^2}$.

---

As $\mathcal{J}(\mathbf{w}, S)$ is $\lambda$-strongly convex, $\mathcal{J}(\mathbf{w}_2, S) \geq \mathcal{J}(\mathbf{w}_1, S) + \frac{\lambda}{2}\|\mathbf{w}_2 - \mathbf{w}_1\|^2$ so that

$$\|\mathbf{w}_2 - \mathbf{w}_1\| \leq \sqrt{\frac{2 \cdot |\mathcal{J}(\mathbf{w}_2, S) - \mathcal{J}(\mathbf{w}_1, S)|}{\lambda}} \leq \sqrt{\frac{2\alpha}{\lambda}}. \tag{3}$$

Conditioned on $\mathbf{w}_1$, from (3) it follows that $\mathbf{w}_2 - \mathbf{w}_1$ has $\ell_2$-sensitivity of $\sqrt{\frac{8\alpha}{\lambda}}$. Therefore, by the standard analysis of Gaussian mechanism [DR$^+$14], it follows that $(\mathbf{w}_2 - \mathbf{w}_1) + \mathbf{H}$ (with $\mathbf{H}$ sampled as in Step 3 of Algorithm $\mathcal{A}_{\mathsf{ObjP-App}}$) satisfies $(\epsilon/2, \delta/2)$-differential privacy. Therefore by standard composition [DR$^+$14], the tuple $(\mathbf{w}_1, \mathbf{w}_2 - \mathbf{w}_1 + \mathbf{H})$ (and hence $\widehat{\mathbf{w}}$) satisfies $(\epsilon, \delta)$-differential privacy. $\qquad\square$

**Theorem 5.7** (Excess population loss guarantee of $\mathcal{A}_{\mathsf{ObjP-App}}$)**.** *Let $\mathcal{D}$ be any distribution over $\mathcal{Z}$, and let $S \sim \mathcal{D}^n$. Suppose that Assumption 5.1 holds and that $\mathcal{W}$ is $M$-bounded. In Algorithm $\mathcal{A}_{\mathsf{ObjP-App}}$, set $\lambda = \frac{2\,L}{M}\sqrt{\frac{2}{n} + \frac{4\,d\,\log(1/\delta)}{\epsilon^2\,n^2}}$, $\alpha = \frac{M^2\lambda}{n^2}$. Then, we have*

$$\Delta\mathcal{L}\left(\mathcal{A}_{\mathsf{ObjP-App}};\, \mathcal{D}\right) \leq O\left(M\,L \cdot \max\left(\frac{1}{\sqrt{n}},\, \frac{\sqrt{d\,\log(1/\delta)}}{\epsilon\,n}\right)\right).$$

*Proof.* Let $\mathbf{w}_1 = \arg\min_{\mathbf{w} \in \mathcal{W}} \widehat{\mathcal{L}}(\mathbf{w};\, S) + \frac{\langle \mathbf{G}, \mathbf{w} \rangle}{n} + \lambda\|\mathbf{w}\|^2$. For $\widehat{\mathbf{w}}$ defined in Step 3 of $\mathcal{A}_{\mathsf{ObjP-App}}$, notice that using Theorem 5.3, $\Delta\mathcal{L}(\widehat{\mathbf{w}};\, \mathcal{D}) \leq \Delta\mathcal{L}(\mathbf{w}_1;\, \mathcal{D}) + L \cdot \mathbb{E}\left[\|\widehat{\mathbf{w}} - \mathbf{w}_1\|\right] \leq O\left(M\,L \cdot \max\left(\frac{1}{\sqrt{n}},\, \frac{\sqrt{d\,\log(1/\delta)}}{\epsilon\,n}\right)\right) + L \cdot \mathbb{E}\left[\|\mathbf{H}\|\right]$. Now, $\mathbb{E}\left[\|\mathbf{H}\|\right] = O\left(\sqrt{\frac{d\alpha\,\log(1/\delta)}{\lambda\epsilon^2}}\right) = O\left(M\,L \cdot \frac{\sqrt{d\log(1/\delta)}}{\epsilon n}\right)$ when $\alpha = \frac{M^2\lambda}{n^2}$. Therefore, $\Delta\mathcal{L}(\widehat{\mathbf{w}};\, \mathcal{D}) \leq O\left(M\,L \cdot \max\left(\frac{1}{\sqrt{n}},\, \frac{\sqrt{d\,\log(1/\delta)}}{\epsilon\,n}\right)\right)$, which completes the proof. $\qquad\square$

*Oracle complexity:* The population loss guarantee of Algorithm $\mathcal{A}_{\mathsf{ObjP-App}}$ is independent of the choice of the exact optimizer $\mathcal{O}$, as long it produces a $\widehat{\mathbf{w}} \in \mathcal{W}$ for an objective function $\mathcal{J}$ such that $\left[\mathcal{J}(\widehat{\mathbf{w}}) - \min_{\mathbf{w} \in \mathcal{W}} \mathcal{J}(\mathbf{w})\right] \leq \alpha$, where $\alpha = \frac{M^2\lambda}{n^2}$ (defined in Theorem 5.7). We will now show that if one uses SVRG (Stochastic Variance Reduced Gradient Descent Optimizer) from [JZ13, XZ14, Bub15] as the optimizer $\mathcal{O}$, then one can achieve an error of $\alpha$ in $O\left((n + \beta/\lambda)\log(1/\alpha)\right)$ calls to the gradients of $\ell(\cdot, \cdot)$, for any $\alpha \in (0, 1]$. The following theorem immediately gives this. Plugging in the value of $\alpha$ from Theorem 5.7, noticing from Theorem 5.2 that $\beta/\lambda \leq \epsilon n$, and considering $\epsilon, M$ and $L$ to be constants, we get the oracle complexity of Algorithm $\mathcal{A}_{\mathsf{ObjP-App}}$ to be $O(n\log(n))$.

**Theorem 5.8** (Convergence of SVRG [JZ13, XZ14, Bub15])**.** *Let $f_1, \cdots, f_n$ be $\beta$-smooth, $\lambda$-strongly convex functions over $\mathcal{W}$, and $\mathcal{F}(\mathbf{w}) = \frac{1}{n} \sum_{i=1}^{n} f_i(\mathbf{w})$. Let $\mathbf{y}^{(1)} \in \mathcal{W}$ be an arbitrary initial point. For $t = \{1, 2, \cdots\}$, let $\mathbf{w}_1^{(t)} = \mathbf{y}^{(1)}$. For $s \in [k]$, let*

$$\mathbf{w}_{s+1}^{(t)} = \mathsf{Proj}_{\mathcal{W}} \left[ \mathbf{w}_s^{(t)} - \frac{1}{10\beta} \left( \triangledown f_{i_s^{(t)}} \left( \mathbf{w}_s^{(t)} \right) - \triangledown f_{i_s^{(t)}} \left( \mathbf{y}_s^{(t)} \right) + \triangledown \mathcal{F} \left( \mathbf{y}^{(t)} \right) \right) \right],$$

*where $i_s^{(t)}$ is drawn uniformly at random from $[n]$, and $y^{(t+1)} = \frac{1}{k} \sum_{s=1}^{k} \mathbf{w}_s^{(t)}$. Then, for $k = 20\beta/\lambda$ the following is true:*

$$\mathbb{E} \left[ \mathcal{F} \left( \mathbf{y}^{(t+1)} \right) \right] - \mathcal{F}(\mathbf{w}^*) \leq 0.9^t \left( \mathcal{F} \left( \mathbf{y}^{(1)} \right) - \mathcal{F}(\mathbf{w}^*) \right).$$

## Footnotes

[1]In this Introduction, we will primarily be concerned with the dependence on $d$ and $n$, and on $(\epsilon, \delta)$-DP. We therefore suppress the dependence on $\delta$ as well as on parameters of the loss function such as Lipschitz constant and the constraint set diameter.

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

# A Proof of Lemma 3.4

Consider $T$ iterations of $\mathcal{A}_{\mathsf{NSGD}}$. Let $\mathbf{G}_1, \ldots, \mathbf{G}_T$ denote the noise vectors and $\mathcal{I}_1, \ldots, \mathcal{I}_T \in [n]^m$ denote the *index* sets of the mini-batches selected in the $T$ iterations. Consider any pair of datasets $S = (z_1, \ldots, z_k, \ldots, z_n)$ and $S' = (z_1, \ldots, z_k', \ldots, z_n)$ differing in exactly one data point $z_k \neq z_k'$ for some fixed $k \in [n]$. Let $\mathbf{w}_0, \mathbf{w}_1, \ldots, \mathbf{w}_T$ and $\mathbf{w}_0, \mathbf{w}_1', \ldots, \mathbf{w}_T'$ denote the trajectories of $\mathcal{A}_{\mathsf{NSGD}}$ corresponding to input datasets $S$ and $S'$, respectively. For any $t \in [T]$, let $\xi_t \triangleq \mathbf{w}_t - \mathbf{w}_t'$.

We follow the proof technique of [FV19, Lemma 4.3]. We prove the following claim via induction on $t$:

$$\mathbb{E}\left[\|\xi_t\|\right] \leq 4\,c\,\frac{\eta\,t}{n},$$

where the expectation is taken over $\mathcal{I}_0, \ldots, \mathcal{I}_{t-1}, \mathbf{G}_0, \ldots, \mathbf{G}_{t-1}$. First, it's trivial to see that the claim is true for $t = 0$. Suppose the claim holds for all $t \leq \tau$. Fix the randomness in $\mathbf{G}_\tau$ and $\mathcal{I}_\tau$. Let $r$ denote the number of occurrences of the index $k$ (where $S$ and $S'$ differ) in $\mathcal{I}_\tau$. By the non-expansiveness property of the gradient update step, we have

$$\|\xi_{\tau+1}\| \leq \|\xi_\tau\| + 4\,c\,\eta\,\frac{r}{m}$$

Now, we now invoke the randomness in $\mathbf{G}_\tau$ and $\mathcal{I}_\tau$. Note that $r$ is a Binomial random variable with mean $m/n$. Hence, by taking expectation and using the induction hypothesis, we end up with

$$\mathop{\mathbb{E}}_{\substack{\mathcal{I}_0, \ldots, \mathcal{I}_\tau \\ \mathbf{G}_0, \ldots, \mathbf{G}_\tau}}\left[\|\xi_{\tau+1}\|\right] \leq 4\,c\,\frac{\eta\,(\tau+1)}{n}$$

This proves the claim. Now, let $\overline{\mathbf{w}}_T = \frac{1}{T}\sum_{t=1}^T \mathbf{w}_t$ and $\overline{\mathbf{w}}_T' = \frac{1}{T}\sum_{t=1}^T \mathbf{w}_T'$. Since $\ell$ is $c$-Lipschitz, thus for every $z \in \mathcal{Z}$, we have

$$\mathop{\mathbb{E}}_{\substack{\mathcal{I}_0, \ldots, \mathcal{I}_{t-1} \\ \mathbf{G}_0, \ldots, \mathbf{G}_{t-1}}}\left[\ell(\overline{\mathbf{w}}_T,\,z) - \ell(\overline{\mathbf{w}}_T',\,z)\right] \leq c \mathop{\mathbb{E}}_{\substack{\mathcal{I}_0, \ldots, \mathcal{I}_{t-1} \\ \mathbf{G}_0, \ldots, \mathbf{G}_{t-1}}}\left[\|\overline{\mathbf{w}}_T - \overline{\mathbf{w}}_T'\|\right] \leq c\,\frac{1}{T}\sum_{t=1}^T \mathop{\mathbb{E}}_{\mathcal{I}_t, \mathbf{G}_t}\left[\|\xi_t\|\right]$$
$$\leq 4\,c^2\,\frac{\eta}{n\,T}\frac{T(T+1)}{2} = 2\,c^2\,\frac{\eta\,(T+1)}{n}$$

This completes the proof.

# B  Proof of Lipschitz property of Moreau envelope (in Part 1, Lemma 4.3)

Fix any $\mathbf{w} \in \mathcal{W}$. We will show that $\|\nabla f_\beta(\mathbf{w})\| \leq 2L$. Define $g(\mathbf{v}) \triangleq f(\mathbf{v}) + \frac{\beta}{2}\|\mathbf{v} - \mathbf{w}\|^2$, $\mathbf{v} \in \mathcal{W}$. Note that $\mathrm{prox}_{f/\beta}(\mathbf{w}) = \arg\min_{\mathbf{v} \in \mathcal{W}} g(\mathbf{v})$. Let $\mathbf{v}^*$ denote $\mathrm{prox}_{f/\beta}(\mathbf{w})$. Now, observe that

$$0 \leq g(\mathbf{w}) - g(\mathbf{v}^*) = f(\mathbf{w}) - f(\mathbf{v}^*) - \frac{\beta}{2}\|\mathbf{w} - \mathbf{v}^*\|^2$$

Thus, we have

$$\frac{\beta}{2}\|\mathbf{w} - \mathbf{v}^*\|^2 \leq f(\mathbf{w}) - f(\mathbf{v}^*) \leq L\|\mathbf{w} - \mathbf{v}^*\|$$

where the last inequality follows from the fact that $f$ is $L$-Lipschitz. Thus, we get $\|\mathbf{w} - \mathbf{v}^*\| \leq 2L/\beta$. By property 3, we have $\|\nabla f_\beta(\mathbf{w})\| = \beta\|\mathbf{w} - \mathbf{v}^*\|$. This together with the above bound gives the desired result.

# C  Optimality of Our Bounds

Our upper bounds in Sections 3 and 4 are tight (up to logarithmic factors in $1/\delta$). In particular, our bounds match a lower bound of $\Omega\left(ML \cdot \max\left(\frac{1}{\sqrt{n}}, \frac{\sqrt{d}}{n}\right)\right)$ on the excess population loss. Such lower can be obtained by combining a lower bound implied by the results of [BST14] together with the known lower bound in the non-private setting. The first term is simply the known lower bound on the excess population loss in the non-private setting. The second term follows from the lower bound in [BST14] on excess empirical loss, and the fact that a lower bound on excess empirical loss implies nearly the same lower bound on the excess population loss. We elaborate on this below.

**Reduction from Private ERM to Private SCO:** To show that a lower bound on the excess empirical loss in [BST14] implies essentially the same lower bound on the excess population loss, it suffices to show the following reduction. For any $\gamma > 0$, suppose there is $\left(\frac{\epsilon}{4\log(2/\delta)}, \frac{e^{-\epsilon}\delta}{8\log(2/\delta)}\right)$-differentially private algorithm $\mathcal{A}$ such that for any distribution on a domain $\mathcal{Z}$, when $\mathcal{A}$ is given a sample $T \sim \mathcal{D}^n$, it yields expected excess population loss $\Delta\mathcal{L}(\mathcal{A}; \mathcal{D}) \leq \gamma$. Then, there is $(\epsilon, \delta)$-differentially private algorithm $\mathcal{B}$ that when given any dataset $S \in \mathcal{Z}^n$, it yields expected excess empirical loss $\Delta\widehat{\mathcal{L}}(\mathcal{B}; S) \triangleq \mathbb{E}_{\mathcal{B}}\left[\widehat{\mathcal{L}}(\mathcal{B}(S); S)\right] - \min_{\mathbf{w}} \widehat{\mathcal{L}}(\mathbf{w}; S) \leq \gamma$.

Fix any $\gamma > 0$. Suppose algorithm $\mathcal{A}$ described above exists. We construct algorithm $\mathcal{B}$ as follows:

1. Given input dataset $S \in \mathcal{Z}^n$, let $\mathcal{D}_S$ be the empirical distribution induced by $S$.

2. Sample $T \sim \mathcal{D}_S^n$.

3. Return $\mathcal{A}(T)$

First, note that $\Delta\widehat{\mathcal{L}}(\mathcal{B}; S) \leq \gamma$. This easily follows from the fact that for any $\mathbf{w}$, $\mathcal{L}(\mathbf{w}; \mathcal{D}_S) = \widehat{\mathcal{L}}(\mathbf{w}; S)$. In particular, observe that

$$\mathbb{E}_{\mathcal{B}}\left[\widehat{\mathcal{L}}(\mathcal{B}(S); S)\right] - \min_{\mathbf{w}} \widehat{\mathcal{L}}(\mathbf{w}; S) = \mathbb{E}_{T \sim \mathcal{D}_S^n, \mathcal{A}}[\mathcal{L}(\mathcal{A}(T); \mathcal{D}_S)] - \min_{\mathbf{w}} \mathcal{L}(\mathbf{w}; \mathcal{D}_S)$$

$$= \Delta\mathcal{L}(\mathcal{A}; \mathcal{D}_S) \leq \gamma.$$

Next, we show that $\mathcal{B}$ is $(\epsilon, \delta)$-differentially private. Let $S = (z_1, \ldots, z_k, \ldots, z_n)$, $S' = (z_1, \ldots, z'_k, \ldots, z_n)$ be neighboring datasets differing in single point whose index is $k \in [n]$. Let $T, T'$ be the samples obtained

by running $\mathcal{B}$ on $S, S'$, respectively, with the same set of random coins in Step 2. More precisely, let $R$ denote the random sampling procedure used in Step 2, and define $T = R(S)$ and $T' = R(S')$. Let $r$ be the number of times the $k$-th point of the input dataset is sampled by $R$. Hence, $r = |T \Delta T'|$, i.e., $r$ is the number of points where $T$ and $T'$ differ. By Chernoff's bound, $r \le 4 \log(2/\delta)$ with probability $1 - \delta/2$. Let $\mathcal{V}$ be any measurable subset of the range of $\mathcal{B}$. Observe that

$$
\begin{aligned}
\mathbb{P}_{\mathcal{B}}[\mathcal{B}(S) \in \mathcal{V}] &= \mathbb{P}_{\mathcal{A},R}[\mathcal{A}(T) \in \mathcal{V}] \\
&\le \mathbb{P}_{\mathcal{A},R}[\mathcal{A}(T) \in \mathcal{V} \mid r \le 4 \log(2/\delta)] \cdot \mathbb{P}[\, r \le 4 \log(2/\delta)] + \delta/2 \\
&\le e^{\frac{r\,\epsilon}{4 \log(2/\delta)}} \cdot \mathbb{P}_{\mathcal{A},R}\left[\mathcal{A}(T') \in \mathcal{V} \mid r \le 4 \log(2/\delta)\right] \cdot \mathbb{P}[\, r \le 4 \log(2/\delta)] + \frac{\delta}{2} + r\, e^{\frac{r\,\epsilon}{4 \log(2/\delta)}} \frac{e^{-\epsilon}\delta}{8 \log(2/\delta)} \\
&\le e^{\epsilon} \cdot \mathbb{P}_{\mathcal{A},R}\left[\mathcal{A}(T') \in \mathcal{V}\right] + \delta \\
&= e^{\epsilon} \cdot \mathbb{P}_{\mathcal{B}}\left[\mathcal{B}(S') \in \mathcal{V}\right] + \delta,
\end{aligned}
$$

where the third inequality follows from the fact that $\mathcal{A}$ is $\left(\frac{\epsilon}{4 \log(2/\delta)}, \frac{\delta}{2}\right)$-differentially private and the notion of group differential privacy. This shows that $\mathcal{B}$ is $(\epsilon, \delta)$-differentially private, proving the reduction, and hence, the lower bound.