[Reviews · NeurIPS 2019]

Reviewer 1



Strengths All the contributions mentioned above. Very clearly written complete work which proves the optimal convergence rate for various different settings. Weaknesses/comments In the abstract it says that "... in the parameter regime most common in practice." Yet, in Algorithm 1 there are the restrictions epsilon \leq 1 and delta \leq 1/n^2. Especially the restriction on delta seems crude. I cannot see where these restrictions come from. The DP result is based here on the reference Abadi et al., and I do not see such restrictions in the results of the reference article. The privacy analysis (and therefor also the convergence bound) of Algorithm 1 seems to be ultimately based on Lemma 3 of [ACG+16] and on the correct asymptotics that its accurate bounds provide. There seems to be an assumption \sigma \geq 1 in Lemma 3 of [ACG+16], however I do not see this appearing in the article. Could you comment on this? Some comment about how the coefficients c_1 and c_2 of Theorem 1 of the reference [ACG+ 16] are determined would be in order. I assume the authors have worked out the result of that Thm. 1 of [ACG+16] in detail and that is where the choices of variance and mini-batch size in Algorithm 1 come from. To be precise, the results of [ACG+16] hold in the case, where the datasets S and S' are such neighbours that one of them is obtained from the other one by removing one element (see e.g. the proof of Lemma 3 of [ACG+16]). Therefore, is it not entirely correct to say (as in Definition 1) "...for any pair of datasets S and S′ differ in exactly one data point..". This gives the impression that one could replace one element in S by another one to obtain S'. Then the results of [ACG+16] would not hold. Perhaps you could comment on the the neighbouring relation of S and S'. Originality Regarding the breadth of the analysis, this is highly original. Quality and clarity Very clearly structured and easy to read, of highest quality. EDIT: Thank you for the response. The authors have answered my comments. I still slightly disagree with one point: I think that from (eps,delta)-DP with "remove"-relation it does not follow (2*eps,2*delta)-DP for the "replace"-relation DP. Using a triangle-like argument, one gets (2*eps,(1+exp(eps))*delta)-DP. Of course that can be bounded with your bound for eps. I suggest you either comment the relation or change the constants in step 1 of Alg. 1 and step 1 of Alg. 2. Moreover, I am still slightly puzzled why specifically \delta \leq 1/n^2, why not e.g. \delta << 1/n. Anyhow that is ok as it is I think. I will keep my score and I vote for accept.

Reviewer 2



I really like the first contribution of the paper. However, I still think this work lacks the experimental part. As I know, most of the recent work on the central (\epsilon, \delta) DP-ERM has experimental study such as [1-6]. So I think in order to say that their method of DP-batch SGD, they should provide some experimental study in order to say the improvement. [1] Towards Practical Differentially Private Convex Optimization. [2] Differentially Private Empirical Risk Minimization Revisited: Faster and More General. [3] Privacy-Preserving ERM by Laplacian Smoothing Stochastic Gradient Descent [4] Renyi Differentially Private ERM for Smooth Objectives [5] Distributed Learning without Distress: Privacy-Preserving Empirical Risk Minimization [6] Efficient Private ERM for Smooth Objectives. ------------------------------------ ------------------------------------------------------------------------------------------------- After the rebuttal I change my rate to accept. However, I still want to see some empirical performance of the algorithms.

Reviewer 3



The optimal rate for ERM is not the same as the optimal rate for stochastic convex optimization. Similarly the optimal rates for private erm are not the same as the optimal rates for private SCO. This paper resolves the important question about what the optimal private rate for SCO is, and gives algorithms for matching the upper bound. The main technique in showing the generalization bounds is uniform stability. They extend the uniform stability analysis of SGD to the noisy SGD case to prove that noisy SGD achieves the optimal SCO rate. The non-smooth case is handled by noisy SGD on the smoothed version of the function via the M-Y envelope. Significance: Significant new results -- worthy of publication Originality: Technically the results seem to follow from standard analysis techniques adapted to the private setting Clarity: Very clearly written

[Author Response · NeurIPS 2019]

We thank all the reviewers for their thoughtful reviews and for pointing out some typos in the manuscript.

**Responses to Reviewer-1's comments:**

*"...the restriction on delta seems crude. I cannot see where these restrictions come from...."*

First, we note that our technical results remain the same without this restriction on $\delta$. However, this restriction is made to comply with a standard requirement on $\delta$ in the literature of differential privacy to ensure the robustness of the definition. In particular, it is standard to require $\delta$ to be much smaller than $1/n$ to rule out trivial mechanisms, e.g., the one that selects one individual uniformly at random and publishes her record in the clear (note that such mechanism satisfies $(0, 1/n)$-differential privacy but is blatantly non-private). This requirement on $\delta$ is discussed in several early references on differential privacy including the textbook by Dwork & Roth (towards the end of Sec. 2.3.3) and the survey by Salil Vadhan, and several others. In fact, in some of these references $\delta$ is even assumed to be a negligible function of $n$ (i.e., smaller than the inverse of any polynomial in $n$).

*"There seems to be an assumption $\sigma \geq 1$ in Lemma 3 of [ACG+16], however I do not see this appearing in the article."*

First, we note that the lower bound on $\sigma$ in [ACG+16, Lemma 3] is because the bound on the norm of $f$ in that lemma (which represents the Lipschitz constant) is assumed to be 1. So, when the Lipschitz constant is $L$, then by simple re-normalization it is easy to see that the analogous lower bound on $\sigma$ is $L$. Second, as pointed out in the supplementary document, due to different normalization of the noise in the gradient update step (noise in our case is normalized by the batch size $m$), our setting of $\sigma$ is smaller than that of [ACG+16] by a factor of $m$. Taking these different normalizations into account, we note that the analogous lower bound on $\sigma$ is $L/m$, and it is indeed satisfied in our case. To see this, note that given the setting of $m$ in Step 2 of Algorithm 1 and the fact that $\epsilon \leq 1$, the setting of $\sigma$ in Step 1 implies that $\sigma \geq 2L/m$.

*"...is it not entirely correct to say (as in Definition 1): 'for any pair of datasets $S$ and $S'$ differ in exactly one data point....' Then the results of [ACG+16] would not hold. Perhaps you could comment on the neighbouring relation of $S$ and $S'$."*

The definition of differential privacy with respect to the adjacency notion involving addition/removal of one element in the dataset is equivalent (up to a factor of 2 in the privacy parameters) to the definition w.r.t. the adjacency notion involving replacement of one element. This follows from a simple triangle-inequality style argument since replacing one element $z$ by another element $z'$ can be carried out via a removal step (of $z$) followed by an addition step (of $z'$). Hence, the same techniques and results of [ACG+16] still apply in our case (after renormalizing the privacy parameters by a factor of 2).

**Response to Reviewer-2's comments:**

*"I still think this work lacks the experimental part. As I know, most of the recent work on the central $(\epsilon, \delta)$ DP-ERM has experimental study such as [1-6]. ... should provide some experimental study in order to say the improvement."*

Our work is the first one to focus on differentially private stochastic convex optimization (as opposed to previous works on private empirical risk minimization). Our primary contribution is theoretical and we believe that the fundamental nature of the question we resolve makes the work interesting for the community. At the same time, our analysis concerns standard algorithms such as mini-batch DP SGD and objective perturbation for which one can easily find experimental results in the literature. The only new algorithmic aspect is the use of prox step (to get Moreau-Yosida smoothing). This step is necessary in the worst case but will not make much difference for simple linear models used in most experiments. Thus we do not think that experiments are likely to give additional insights into the question we investigate.

[Meta-Review · NeurIPS 2019]

The author's rebuttal have answered most of the (minor) weak points raised by the reviewers. In particular, this result stands on the theoretical side and even if some experiments could be conducted, they would not significantly change the interest or impact of the result.